# Hydraulic transmissivity inferred from ice-sheet relaxation following Greenland supraglacial lake drainages

Ching-Yao Lai [1,2,3 ✉], Laura A. Stevens [2,4], Danielle L. Chase[5], Timothy T. Creyts[2], Mark D. Behn[6], Sarah B. Das[7] & Howard A. Stone [5]

Surface meltwater reaching the base of the Greenland Ice Sheet transits through drainage networks, modulating the flow of the ice sheet. Dye and gas-tracing studies conducted in the western margin sector of the ice sheet have directly observed drainage efficiency to evolve seasonally along the drainage pathway. However, the local evolution of drainage systems further inland, where ice thicknesses exceed 1000 m, remains largely unknown. Here, we infer drainage system transmissivity based on surface uplift relaxation following rapid lake drainage events. Combining field observations of five lake drainage events with a mathematical model and laboratory experiments, we show that the surface uplift decreases exponentially with time, as the water in the blister formed beneath the drained lake permeates through the subglacial drainage system. This deflation obeys a universal relaxation law with a timescale that reveals hydraulic transmissivity and indicates a two-order-of-magnitude increase in subglacial transmissivity (from $0.8 \pm 0.3\,mm^3$ to $215 \pm 90.2\,mm^3$) as the melt season progresses, suggesting significant changes in basal hydrology beneath the lakes driven by seasonal meltwater input.

[1] Department of Geosciences, Princeton University, Princeton, NJ, USA. [2] Lamont-Doherty Earth Observatory, Columbia University, Palisades, NY, USA. [3] Program in Atmospheric and Oceanic Sciences, Princeton University, Princeton, NJ, USA. [4] Department of Earth Sciences, University of Oxford, Oxford, UK. [5] Department of Mechanical and Aerospace Engineering, Princeton University, Princeton, NJ, USA. [6] Department of Earth and Environmental Sciences, Boston College, Chestnut Hill, MA, USA. [7] Department of Geology and Geophysics, Woods Hole Oceanographic Institution, Woods Hole, MA, USA. ✉email: cylai@princeton.edu

Thousands of supraglacial lakes form annually on the surface of the Greenland Ice Sheet in response to seasonal melt that fills topographic depressions. Many of these lakes drain rapidly (<1 day), transporting meltwater through vertical hydrofractures directly to the ice-bed interface[1–3]. Surface meltwater entering the subglacial drainage system at the ice-bed interface plays crucial roles in modulating the flow of the ice sheet[1,2,4–7]. As the melt season progresses, observations and theories suggest that the basal hydrologic system beneath the ice sheet transitions from an inefficient distributed network to a more efficient channelized system that transmits more water at lower pressure[8–11]. Drainage type, whether distributed or channelized, impacts basal sliding and ice discharge[5,8–10].

Understanding the degree and the spatial extent of the drainage system transition is paramount for characterizing the relationship between seasonal ice flow perturbations and surface melt. Field-based tracer studies conducted in the western margin of the ice sheet[10,11] suggest a progressive channelization throughout the melt season, and indicate that a 5-fold increase in subglacial water speed occurs as a channelized system develops near the terminus[10]. However, the effect of channelization tapers further inland[12] at higher elevations (roughly > 1000 m above sea level [a.s.l.])[13,14], where evolution of local basal hydrology remains poorly constrained by observations. Furthermore, observational-based estimates for hydraulic transmissivity, a key parameter controlling the water discharge in a subglacial sheet for a given hydraulic potential gradient, are scarce. Here we calculate the local hydraulic transmissivity[9,15] beneath rapidly draining supraglacial lakes (1000–1350 m a.s.l.) at their time of drainage using a combination of field observations, a mathematical model, and laboratory experiments in order to quantify the seasonal changes of local hydraulic transmissivity driven by seasonal meltwater inputs.

## Results

**Relaxation of surface uplift following lake drainage.** The rapid drainage of lakes injects surface meltwater into the ice–bed interface. The water can be transiently stored in water-filled "blisters" (Fig. 1a) that lift and deform the overlying ice sheet[2,3,6,16]. After a lake drainage event, surface uplift decays over time[1] as water exits the blister and spreads along the ice–bed interface (Fig. 1b). Previous studies have documented the initial stages of lake drainage and blister formation[1,2] and modeled the effect of lake drainage on the subglacial hydrological system[13]. Here we focus on the long-term surface relaxation (1–10 days) of the ice sheet after a lake drains to quantify the hydraulic transmissivity[9,15] of the surrounding subglacial drainage system.

Using ice-sheet surface elevation data from on-ice GPS stations, we characterize the relaxation timescale for five lake drainage events that occurred between 2006 and 2012 at three separate lakes located up to ~200 km apart along the western margin of the Greenland Ice Sheet (Fig. 1d–h; Supplementary Table 1). The five rapid lake drainages have drainage dates that span from the early melt season (typically May through June) through the mid-melt season (typically July through August). We find a wide range of relaxation timescales of surface uplift (from 12 h to 10 days) depending on when lake drainage occurs within the melt season (Fig. 1d–h). Ice-sheet surface uplift relaxation is fitted by an exponential function (dashed curves in Fig. 1d–h and Supplementary Fig. 4f–j) to quantify the relaxation time $t_{rel}$. The two uplift peaks following the 2011 drainage shown in Fig. 1e likely result from additional water injection into the blister from nearby surface or basal sources (Methods). We set time zero to be the time at which uplift begins to relax continuously back to pre-drainage values. Longer relaxation times $t_{rel}$ are generally observed for drainages that occur earlier in the melt season (e.g., North Lake 2011 (Fig. 1e)) and shorter $t_{rel}$ are observed for drainages later in the melt season (e.g., North Lake 2006 (Fig. 1h)).

**Blister relaxation model.** Relaxation of surface uplift following a lake drainage event can be explained by gradual drainage of the blister into the neighboring subglacial drainage system (Fig. 1b). Based on this conceptual model we hypothesize that the relaxation timescales are controlled by the subglacial drainage system beneath the ice sheet. To test this hypothesis, we develop a model that links the surface relaxation to the hydraulic transmissivity of the drainage system.

Near the ice-sheet margin the subglacial drainage system is thought to evolve seasonally[5,8–11] from an inefficient, distributed drainage system early in the melt season to an efficient, channelized system later in the melt season. However, at elevations where supraglacial lakes form (~1000 m a.s.l.), field evidence suggests that channels do not develop until early August[10]. Subglacial hydrological models indicate that short, discontinuous channels form following lake drainage, but the discharge from the channel is small compared to the discharge through the distributed sheet[13]. In reality, either a purely distributed system without any channels or a channelized system dominated by turbulent discharge are unlikely end members of the subglacial drainage system beneath the lakes. Therefore, instead of quantifying the efficiency of a purely distributed sheet or a purely turbulent channel, here we use transmissivity to characterize the bulk subglacial drainage system (valid within 2–6 km horizontal distance from the lake, as explained below), where a wide range of drainage features likely co-exist (e.g., distributed flows through porous sediment sheets[17], thin films[18–20] linked cavities[21,22], localized flows through channels, and weakly connected systems[23]). A similar approach was developed by Sommers et al.[24]. Because the horizontal extent of the water flow (~kilometers) is much larger than the vertical extent (~meters), the bulk subglacial drainage system is treated as a continuum water sheet of effective depth $h_0$, effective permeability $k$, and porosity $\phi$ (ratio between water-filled space to total space).

Uplift relaxation is driven by the elastic deformation of ice lying above the water-filled blister and resisted by the viscous dissipation of the water flow in the distributed drainage system. Below we use scaling arguments to obtain a characteristic timescale of blister relaxation $t_c$. The full mathematical model for blister relaxation dynamics is detailed in Supplementary Information Section 1.

For a blister of maximum thickness $H$ and radius $R$ (Fig. 2a) the blister volume scales as $V \approx 2\pi\alpha H R^2$ (equation (S.15)), where $\alpha$ is a dimensionless parameter related to the shape of the blister. From mass conservation, the blister volume relaxation rate ($dV/dt \approx V/t_c$) equals the water flux entering the subglacial water sheet with flow velocity $u_p$ (red arrows in Fig. 2a) at radial distance $r$. Thus, a mass balance yields (see equations (S.12) and (S.15))

$$\frac{V}{t_c} \approx \frac{2\pi\alpha H R^2}{t_c} \approx r\phi u_p h_0 \qquad (1)$$

The flow in the bulk subglacial water sheet can be described by Darcy's law for flow in porous materials[17–22], where the water flux $\phi\boldsymbol{u_p}$ (volume per time per unit area crossing the flow) increases linearly with the gradient of water pressure $p$. Assuming

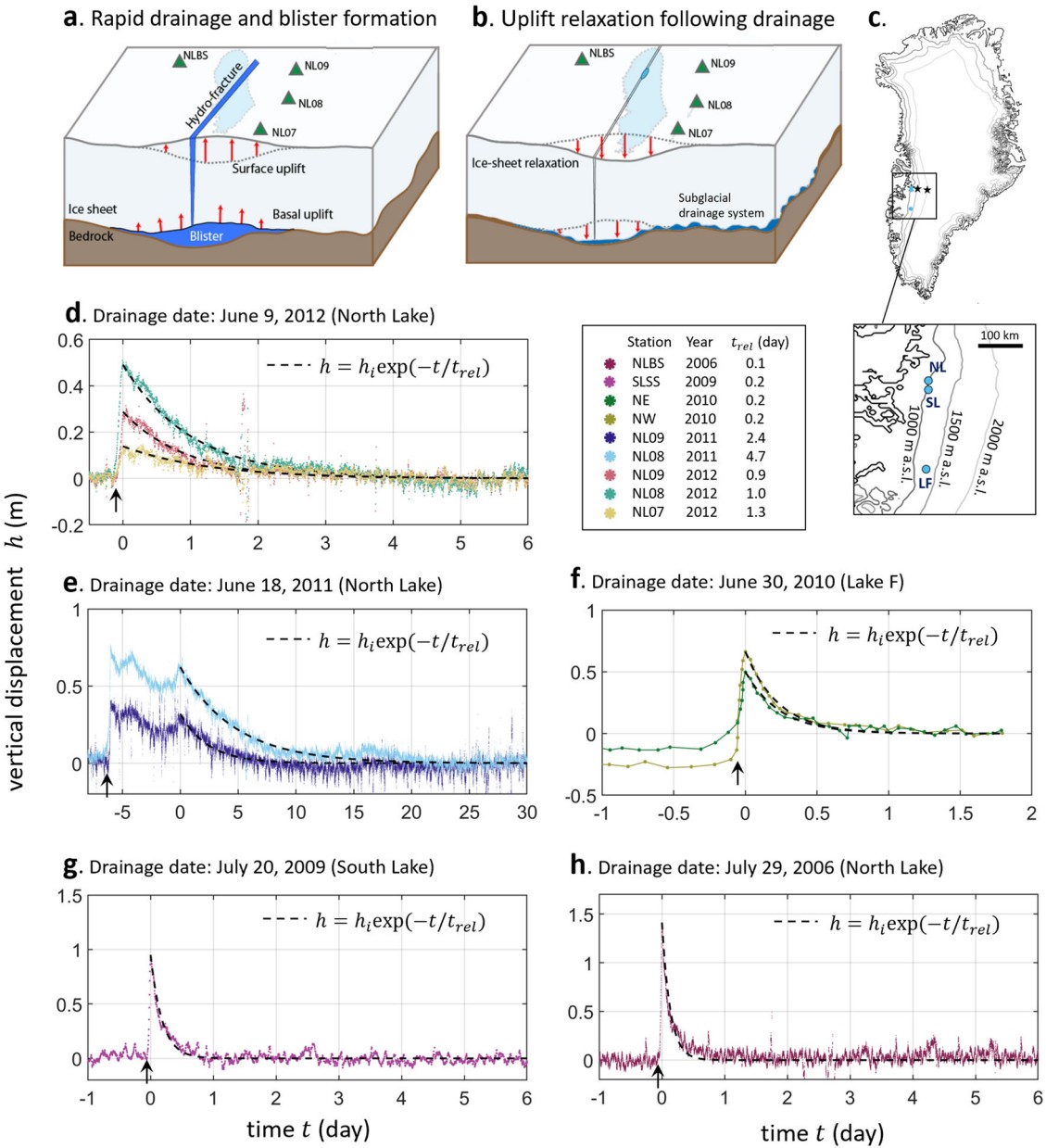

**Fig. 1 Ice-sheet uplift relaxation following rapid supraglacial lake drainage.** Schematic drawings of the North Lake GPS array that reports ice-sheet surface uplift and speed (**a**) during subglacial blister formation at the time of rapid drainage, and (**b**) post-drainage relaxation as the blister drains into the surrounding ice-bed interface. **c** The locations of North Lake (NL), South Lake (SL), and Lake F (LF) are marked by the blue dots. **d**–**h** Ice-sheet surface elevation during rapid drainage (drainage start time marked by arrows) and post-drainage uplift relaxation from GPS stations for five different drainages, where post-drainage uplift relaxation begins at time = 0 days. Dashed curves show vertical displacement data fit to the exponential function, $h(t) = h_i \exp(-t/t_{rel})$. The fitted $t_{rel}$, the lake drainage year, and the station name for each GPS station are listed in the table. For the 2011 data, we therefore set time zero to be the time after which uplift only relaxes, and no significant amounts of additional water enter the blister (Methods).

the discharge in channels are negligibly small compared with that of the bulk subglacial system[13], Darcy's law holds to the first order. Thus, the bulk subglacial flow obeys Darcy's law $\phi \boldsymbol{u_p} = -\frac{k}{\mu} \nabla p$, where $\mu$ is water viscosity. Note that in both field data and our laboratory experiments the blister radius $R$ remains unchanged during blister relaxation (Fig. 2b) and is assumed constant in our model. We estimate that the horizontal viscous pressure drops in the water sheet and the blister for $r < R$ are negligibly small (Supplementary Information Section 1.5) compared with the viscous pressure drop $\Delta p_v$ in the water sheet at $r > R$. Integrating the pressure gradient radially along the water

sheet over $r > R$ and considering mass conservation, we obtain a horizontal pressure drop (equation (S.18)) of magnitude.

$$\Delta p_v \approx \frac{\mu}{h_0 k} \frac{V}{t_c} \ln(\frac{R_p}{R}) \qquad (2)$$

where $R_p$ is the radius of the invading water front in the water sheet.

Deformation of the ice overlying the blister generates elastic stresses in the ice sheet. For a penny-shaped blister[3,25–27], the magnitude of the elastic stresses in ice can be estimated by Hooke's law corresponding to an approximate strain $H/R$

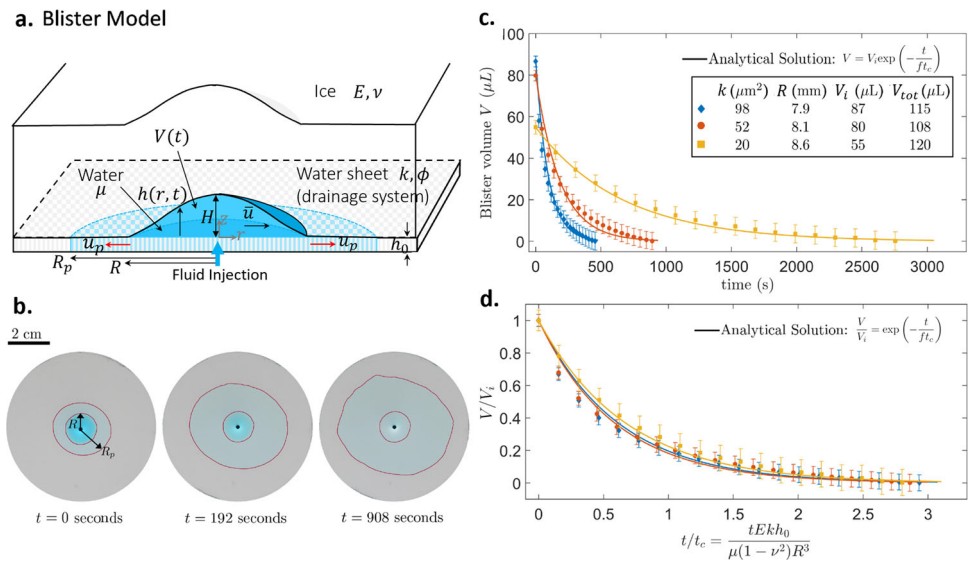

**Fig. 2 Experimental validation of the mathematical model. a** Schematic of the blister model. An elastic layer (ice) with Young's modulus $E$ and thickness $d$ over a porous substrate (drainage system) of thickness $h_0$, porosity $\phi$, and permeability $k$. Injection of a liquid with volume $V_{tot}$ and viscosity into the interface between the elastic layer and the substrate forms a blister. The experimental parameters are listed in Table 1 and the uncertainties are listed in Supplementary Table 3. **b** The top view of the experimental relaxation dynamics, during which liquid in the blister (dark blue) enters the pore space (light blue), increasing fluid area in the porous substrate. The blister and the fluid front in the porous substrate are outlined. During relaxation the blister radius $R$ remains approximately constant. **c** Measured blister volumes $V(t)$ for three different substrate permeabilities decrease exponentially with time. The analytical solution is given by Eq. (6). **d** The dimensionless experimental data fall onto a common curve, agreeing well with the exponential solution (Eq. (6) with $f = 0.6 - 0.7$ (Table 1); solid curves).

(see equation (S.16)):

$$\Delta p_e \approx \frac{EH}{2(1 - \nu^2)R} \qquad (3)$$

For a typical blister radius $R \approx 2$ km and maximum thickness $H \approx 1$ m, and ice with Young's modulus $E \approx 10$ GPa and Poisson's ratio $\nu \approx 0.3$, the elastic stress is $\Delta p_e \approx 2$ MPa. The elastic stresses caused by the blister are balanced by an increase in water pressure in addition to the hydraulic potential. Thus, the water pressure in the blister at the base of a uniform ice sheet of thickness $d \approx 1$ km is $p = \rho_i g d + \rho_w g h + \Delta p_e$ (equation (S.38)), where $h(r, t)$ is the water thickness in the blister, $\rho_w$ and $\rho_i$ are the water and ice density, respectively, and $g$ is the gravitational acceleration. Subglacial flow is driven by the water pressure gradient $\nabla p$, where the gradient of ice and water overburden pressure are negligible compared with that of the elastic stress (Supplementary Information Section 2). Thus, during ice-sheet relaxation the elastic stresses, rather than the ice and water overburden pressure, dominantly drive subglacial flow and blister relaxation.

The relaxation dynamics are governed by a balance between the elastic stresses $\Delta p_e$ driving the relaxation and the pressure drop $\Delta p_v$ resisting the subglacial flow, i.e., $\Delta p_e \approx \Delta p_v$. Thus, Eqs. (1–3) give the characteristic relaxation time $t_c = 4\pi\alpha\ln(R_p/R)\mu R^3(1 - \nu^2)/(Ekh_0)$. Neglecting the constants of magnitude order one (e.g., $\pi$) and the numerical pre-factor $\alpha\ln\left(\frac{R_p}{R}\right) \approx f = O(1)$ (equation (S.26)) that do not affect the scaling, we obtain the characteristic relaxation time:

$$t_c = \mu R^3(1 - \nu^2)/(Ekh_0) \qquad (4)$$

This timescale will be used to rescale the experimental and field data.

Considering mass conservation and a force balance analysis (Supplementary Information Section 1), we obtain a nonlinear

ordinary differential equation for the blister volume $V(t)$:

$$\frac{E}{1(1 - \nu^2)}\frac{V}{2\alpha\pi R^3} + \frac{\mu}{4\pi h_0 k}\ln\left(\frac{V_{tot} - V}{\phi\pi h_0 R^2}\right)\frac{dV}{dt} = 0 \qquad (5)$$

where $V_{tot}$ is the total volume of water in the system. Eq. (5) can be approximated by a linear ordinary differential equation (Supplementary Information Section 1.4; equation (S.25)), yielding exponential solutions for the blister volume $V(t)$ and thickness $h(r, t)$ as a function of time $t$ and radial distance $r$:

$$V(t) = V_i\exp\left(-\frac{t}{ft_c}\right) \qquad (6)$$

$$h(r, t) = h_i(r)\exp\left(-\frac{t}{ft_c}\right) \qquad (7)$$

where $V_i = V(t = 0)$, $h_i(r) = h(r, t = 0)$, and $f$ is a numerical pre-factor (equation (S26)). Comparing Eq. (7) with $h(t) = h_i\exp(-t/t_{rel})$ gives $f = t_{rel}/t_c$. The analytic approximation is in close agreement to the numerical solution (Supplementary Fig. 1). The relaxation dynamics are negligibly impacted by the bed slope (Supplementary Information Section 2) and the melting and viscous motion of ice (Supplementary Information Section 3).

Based on field observations and lab experiments, we assume $R$ is constant so that the only way for blister volume to decrease is by pushing water through the porous water sheet, yielding an exponential decay of blister height. In contrast, Hewitt et al.[16] considered a blister propagating on a water-filled porous sheet with an increasing $R$ and decreasing $h$. In their model, water volume in the blister $V$ is constant and does not leak into the porous water sheet, resulting in a power-law decrease of blister height as a function of time[16]. A comparison of the GPS uplift data with the power-law decay[16] and our exponential decay (Eq. (7)) is shown in Supplementary Fig. 7. The GPS uplift data exhibits better agreements with our exponential solutions.

**Laboratory experiments.** Next, we tested the analytic model (Eqs. (4) and (6)) against our laboratory experiments, which allow precise control and direct measurement of sheet permeability. Experiments on fluid peeling between an elastic sheet and a non-permeable rigid substrate have previously been investigated[28]. In contrast, here we consider a porous substrate that mimics the drainage system. In our experiment, a fluid-filled blister (dark blue in Fig. 2b) is generated via liquid injection into the interface between a transparent elastic layer and a porous substrate. This setup mimics ice lying above the drainage system (Fig. 2a). After injection of liquid, the fluid permeates from the blister through the porous substrate (light blue in Fig. 2b), the blister thickness decreases and the blister radius remains unchanged (Supplementary Movie 1), which differs from the increasing $R$ in Hewitt et al. (2018)[16]. Our lab experiments show that varying the permeability of the porous substrate $k$ significantly impacts the relaxation timescale in the experiments (Fig. 2c). The blister volume $V(t)$ relaxes exponentially with time, validating our analytical solution (Eq.(6), curves in Fig. 2c). All parameters in the analytical solution can be calculated based on the experimental variables except for the pre-factor $f$ (Table 1). Since $t_c \propto R^3$ via Eq.(4), the relaxation time ($\sim10^3$ seconds) of a laboratory-scale blister ($R\sim10$ mm) is expected to be much shorter than that observed in the field. After rescaling the blister volume $V$ with $V_i$ and time $t$ with $t_c$ (Eq.(4)), the experimental data for different permeabilities collapse onto a universal curve (Fig. 2d), demonstrating the model's success in predicting the impact of permeability on the relaxation timescale. Here the thickness $d$ of the overlying elastic layer is roughly the same as the blister radius $R$, similar to field observations. When $d \ll R$ the relaxation timescale obeys a different scaling[29].

**Inferring hydraulic transmissivity from surface GPS data.** We then applied the validated model to estimate hydraulic transmissivity $kh_0$[15] of the subglacial water sheet from surface uplift data following the five lake drainage events (Supplementary Table 1). The relaxation time $t_{rel}$ of each set of GPS data was obtained by fitting $h(t) = h_i\exp(-t/t_{rel})$ (dashed curves in Fig. 1) to the GPS data. According to Eq.(4) and $f = t_{rel}/t_c$, the hydraulic transmissivity can be expressed as $kh_0 = \frac{\mu R^3(1-\nu^2)}{Et_c} = f\frac{\mu R^3(1-\nu^2)}{Et_{rel}}$ (lines in Fig. 3a) and calculated using the relaxation time $t_{rel}$, the Young's modulus of ice $E$, water viscosity $\mu$, Poisson's ratio $\nu$, blister radius $R$ estimated from the lake volume $V_{tot}$ (Methods), and the pre-factor $f = O(1)$ (equation (S.26); Table 1). This transmissivity estimate is valid within a 2–6 km horizontal distance (the extent of water in the sheet displaced by the lake volume) from the lake-drainage location. The transmissivity estimated for each lake drainage event based on the GPS uplift data is shown in Fig. 3a and Table 1. Note that the variability of the transmissivity estimated from different stations for the same drainage event is less than that between different drainage events. The uncertainties propagated in the transmissivity estimate are reported in Supplementary Table 2 (Methods). The changes in transmissivity $kh_0$ could be caused by the evolution of not only the effective permeability $k$ (or conductivity $K \equiv \frac{k\rho_w g}{\mu}$[30]) but also effective thickness $h_0$ of the bulk subglacial drainage system.

**Universal uplift relaxation dynamics.** To demonstrate the universality of the time-dependent uplift relaxation data, we rescaled the surface uplift data (Fig. 3b) by the characteristic relaxation time $t_c$, as calculated from Eq. (4) using the parameters listed in Fig. 3a, and the initial vertical displacement $h_i$. The rescaled data collapse (Fig. 3c) onto the analytical solution, Eq. (7) Thus,

**Table 1 Parameters and their definitions used in this study.**

| | Porous sheet | | Elastic layer | | Liquid | | Blister | | | Dimensionless parameters | | | Numerical factors | |
| | φ | $kh_0$ | E | d | μ | ν | R | $V_i$ | $V_{tot}$ | A | B | C | α | f |
|---|---|---|---|---|---|---|---|---|---|---|---|---|---|---|
| Field data | 0.5 | 215.0 mm³ | 10 GPa | 1 km | 1 mPa·s (water) | 0.3 | 3.2 km | 0.040 km³ | 0.044 km³ | $2.2\times10^{-6}$ | 1.14 | 0.04 | 0.32 | 0.79 |
| | | 100.0 mm³ | | | | | 2.9 km | 0.027 km³ | 0.030 km³ | $1.5\times10^{-6}$ | | 0.05 | | 0.75 |
| | | 43.0 mm³ | | | | | 2.4 km | 0.016 km³ | 0.018 km³ | $1.1\times10^{-6}$ | | 0.06 | | 0.70 |
| | | 44.0 mm³ | | | | | 2.4 km | 0.016 km³ | 0.018 km³ | $1.1\times10^{-6}$ | | 0.06 | | 0.70 |
| | | 1.7 mm³ | | | | | 1.8 km | 0.007 km³ | 0.008 km³ | $9.3\times10^{-8}$ | | 0.08 | | 0.61 |
| | | 0.8 mm³ | | | | | 1.8 km | 0.007 km³ | 0.008 km³ | $4.7\times10^{-8}$ | | 0.08 | | 0.61 |
| | | 6.2 mm³ | | | | | 2.2 km | 0.007 km³ | 0.008 km³ | $1.2\times10^{-6}$ | | 0.11 | | 0.48 |
| | | 5.7 mm³ | | | | | 2.2 km | 0.007 km³ | 0.008 km³ | $1.1\times10^{-6}$ | | 0.11 | | 0.48 |
| | | 4.5 mm³ | | | | | 2.2 km | 0.007 km³ | 0.008 km³ | $8.4\times10^{-7}$ | | 0.11 | | 0.48 |
| Lab exp. | 0.5 | 98×90 μm³ | 217 kPa | 1 cm | 0.8 Pa·s | 0.5 | 7.9 mm | 87 μL | 115 μL | $1.6\times10^{-3}$ | 1.32 | 0.10 | 0.32 | 0.61 |
| | | 52×90 μm³ | | | | | 8.1 mm | 80 μL | 108 μL | $1.3\times10^{-3}$ | 1.35 | 0.12 | | 0.58 |
| | | 20×90 μm³ | | | | | 8.6 mm | 55 μL | 120 μL | $2.2\times10^{-3}$ | 2.18 | 0.19 | | 0.67 |

Although the dimensional governing equation (equation (S.21)) depends on nine dimensional parameters ($\phi$, $h_0$, $k$, $E$, $\nu$, $\mu$, $V_{tot}$, $V_i$, $R$), its dimensionless form (equation (S.23)) only depends on two dimensionless parameters ($B$, $C$). We designed the experimental parameters so that the dimensionless parameters ($B$, $C$) of the experiments match that of the field data, meaning experiments fall into the same physical regimes as the field observations. Derivations of the governing equation and the non-dimensionalization are detailed in Supporting Information. Numerical factor $\alpha \approx 0.32$ was found empirically by fitting all experimental data to Eq. (6). When calculating $C$ for the field data, we assumed the thickness of the water sheet $h_0$ is on the order of 0.1 m[13]. Definitions of parameters:
Porous sheet properties: $\phi$: porosity, $h_0$: thickness, $k$: permeability, $kh_0$: transmissivity
Elastic layer properties: $E$: Young's modulus, $d$: thickness, $\nu$: Poisson's ratio
Liquid: $\mu$: viscosity, $V_{tot}$: volume of total injected liquid
Blister: $R$: radius, $V_i$: initial volume
Dimensionless parameters: $A = \frac{k\phi h_0 R^2}{V_i^3}$, $B = \frac{V_{tot}}{V_i}$, $C = \frac{\phi h_0 R^2}{V_i}$, $f = \alpha \ln(\frac{B-\gamma}{C})$, where $\gamma = \frac{e-1}{e} = 0.63$

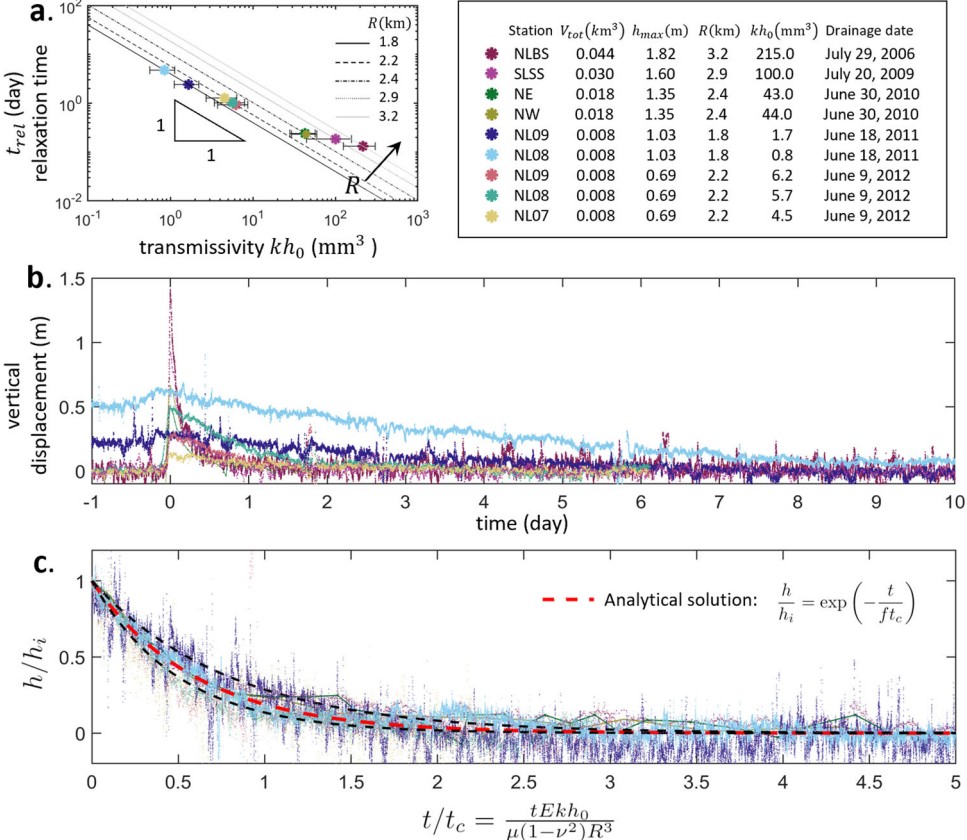

**Fig. 3 Hydraulic transmissivity and the universal dynamics of uplift relaxation. a** The relaxation time $t_{rel}$ obtained from surface uplift data from different stations and the predicted hydraulic transmissivity $kh_0$. Detailed information for each data set is shown in the table. Here the supraglacial lake volume is assumed to be the total volume $V_{tot}$ injected into the blister and the water sheet. The black lines ($t_{rel} = f\frac{\mu R^3(1-\nu^2)}{E}(kh_0)^{-1}$) are the model predictions for different blister radii $R$. The lake volume $V_{tot}$ of Lake F and North Lake listed in the table are taken from references[1, 2, 6] and the lake volume of South Lake is estimated in the Methods. Surface uplift data from five drainages of three different lakes as recorded by seven different GPS stations are plotted in (**b**) dimensional and (**c**) dimensionless forms. Despite a wide range of relaxation times, when rescaled by the characteristic relaxation time $t_c$ (Eq. (4)) and initial vertical displacement $h_i$, all field uplift data collapse onto the exponential analytical solution (Eq. (7), red dashed line (**c**) with $f = 0.6$ averaged over all datasets (Methods)). Upper and lower dashed lines represent the solutions with the highest and lowest $f$ (Table 1), respectively, among all the data sets.

despite variability in local lake basin and ice-sheet bed geometry, surface uplift magnitude, and timescale of uplift relaxation, the collapse of the data onto a universal curve indicates that the relaxation dynamics depends to first order on two dimensionless parameters, $h/h_i$ and $t/t_c$. Notably, both the dimensionless field data (Fig. 3c) and dimensionless experimental data (Fig. 2d) fall onto the exponential solution (Eqs. (6–7)), demonstrating the universality of the uplift relaxation dynamics.

**Seasonal changes of transmissivity under supraglacial lakes.** Finally, we compare the transmissivities for lakes draining during the early- and mid-melt season. Our results suggest the local transmissivity of the basal drainage system beneath North Lake can change by up to two orders of magnitude between early June and late July. Early-season events relax more slowly as characterized by a low-transmissivity drainage system ($kh_0 = O(1)$ mm³). By contrast, the mid-season events relax faster and are best explained by high transmissivity ($kh_0 = O(10^2)$ mm³). When comparing our estimates of hydraulic transmissivity to modeled cumulative surface runoff[31] at the time of lake drainage, we observe that transmissivity generally increases with increased cumulative runoff (Fig. 4a). Our results suggest that local transmissivity beneath North Lake at ~1000 m a.s.l. differs by up to two orders of magnitude between June 9 (2012 drainage) and July

29 (2006 drainage) as the cumulative surface melt volume differs by a factor of 25 (Fig. 4a). The seasonal change of hydraulic conductivity is important for reconciling mismatch between modeled and observed subglacial water pressure[32]. Downs et al (2018)[32] included a simple linear relationship between melt input and hydraulic conductivity in a subglacial hydrological model and found an improved match between the modeled subglacial water pressure with borehole observations in the late melt season. Our work provides observational evidence of the seasonal changes of hydraulic transmissivity. Note that our result does not imply that transmissivity solely depends on cumulative surface runoff. The processes controlling the shift in transmissivity are left for future studies. Comparing the range of transmissivity values we obtained from the field data ($kh_0 = O(1 - 10^2)$ mm³) with the hydraulic conductivity used in large-scale hydrological models[13] ($K \equiv \frac{k\rho_w g}{\mu} \approx 0.05 - 0.5$ m s⁻¹), we calculate that the sheet depth is of order $h_0 = O(10^{-2} - 10^1)$ m. All parameters used in the transmissivity calculation are listed in Table 1.

## Discussion
We infer that local hydraulic transmissivity under North Lake (~1000 m a.s.l.) likely changes seasonally. Our method can be applied to estimate local transmissivity beneath other draining supraglacial lakes simply based on surface observation of uplift

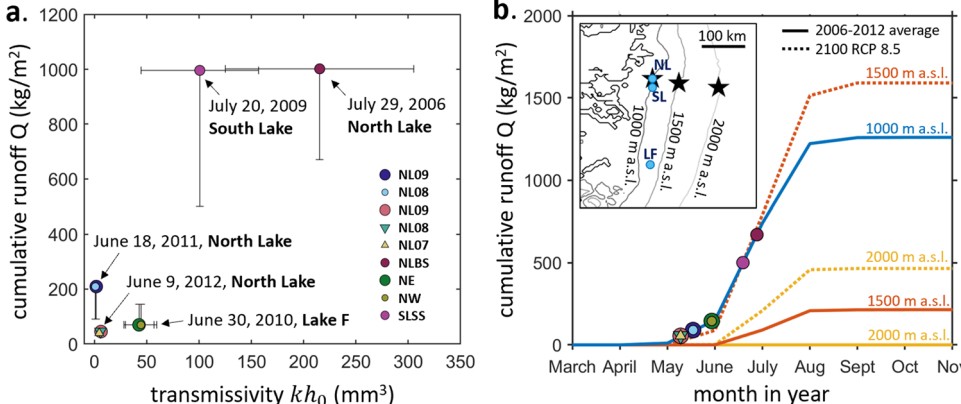

**Fig. 4 Seasonal variation of hydraulic transmissivity. a** Transmissivity inferred from five drainage events across three lakes and the corresponding cumulative runoff at the lake drainage time. Cumulative runoff is the sum of daily, 11-km resolution RACMO runoff estimates[31] from the first day of the year up to the drainage date at the nearest RACMO grid cell to the drainage location (Methods). Vertical error bars (Methods) show the difference in cumulative runoff estimates between RACMO[31] and MAR[34] runoff estimates. **b** Time evolution of cumulative runoff (monthly output) obtained from the regional climate model MAR averaged from 2006 to 2012 (solid curves) and its 2100-projections under the RCP 8.5 scenario[34] (dashed curves), evaluated at the three locations (elevations) marked as stars on the inset map (same map as the inset in Fig. 1c). The star at 1000 m a.s.l. is the North Lake location. The labels mark the day of year (DOY) of the drainage events and the corresponding 2006-2012 averaged cumulative runoff.

and lake volume. Dye and gas tracer experiments[10] have the advantage of tracking subglacial water speed at different times of the year, which well characterize the evolution of subglacial efficiency near the marginal areas. To extend these experiments to higher-elevation regions (>1000 m a.s.l.; far away from the ice-sheet margin), however, the measured water velocity must be averaged along the entire tracer pathway (e.g., the tracing experiment performed at moulin L41 by Chandler et al. (2013)[10] gives a water velocity averaged along a 41-km-long water pathway). Thus, it is unclear to what extent the drainage system evolves locally in higher elevation regions from dye and gas tracer experiments. Our results imply that the drainage system likely evolves locally at the lake elevations (~1000 m a.s.l.). In addition, tracer experiments are limited to sampling moulins in land-terminating glaciers located mainly in south-west Greenland. By contrast, supraglacial lakes are widespread over the ice sheet and are forming in progressively higher elevation regions as the climate warms[33].

Quantifying hydraulic transmissivity of high-elevation areas (>1000 m a.s.l.) is crucial for understanding ice-sheet behavior in a warming climate. Our method provides an approach to determine the magnitude of transmissivity of high-elevation regions using only observations of surface deformation and lake volume. Under RCP scenario 8.5[34], total melt season runoff at a 1500 m a. s.l. location up-flowline of North Lake in 2100 is projected to be higher than the average total melt season runoff from 2006 to 2012 observed at North Lake (~1000 m a.s.l.) (Fig. 4b). Thus, over the coming decades as runoff increases[35] and if surface-to-bed meltwater pathways migrate inland to higher elevations[36,37], we would expect an overall increase in transmissivity in the ice-sheet interior. Such an increase would likely impact the timescale of sliding in response to meltwater accessing the bed[9]. Indeed, we observe transient ice flow velocities to remain elevated over a shorter time period for high-transmissivity drainage systems (Supplementary Fig. 6a, b), compared to a less efficient bed drainage (Supplementary Fig. 6c, d) for the North and South Lake drainage events presented here. To date, our observations of transmissivity are limited to the elevations of the three lakes investigated in this study, which are all located in the southwest to central west sectors of the Greenland Ice Sheet. Observations are needed at higher elevation, rapidly draining lakes located across

all sectors of the ice sheet to constrain processes governing the present and future evolution of sliding in the ice-sheet interior.

## Methods

**GPS data.** For the 2006, 2011, and 2012 North Lake and 2009 South Lake drainage events, continuous 30-s resolution GPS data collected by dual-frequency Trimble NetR9 receivers were processed with Track software[38], following the methodology previously presented for the same GPS data in Stevens et al. [2] and Stevens et al. [39]. GPS data for each station were processed individually relative to the 30-s resolution Greenland GPS Network (GNET) KAGA base station located on bedrock, 55 km from North Lake[40]. 1-$\sigma$ error output from Track software for these data are the order of ± 2 cm in the horizontal and ± 5 cm in the vertical across all stations and years[2]. GPS data are archived at UNAVCO (see Data and materials availability).

**Uplift data processing.** The vertical displacement of each GPS station shows trends related to background ice sheet advection through the lake basins. For example, in Supplementary Fig. 4a–e the background vertical displacement $y(t)$ slowly varies with time $t$ before and after the uplift peak caused by the 5 lake drainage events. These background trends are fitted to a linear model ($y = at + b$, where $a$, $b$ are constants) and subtracted to yield the detrended vertical displacement $h(t)$ (Supplementary Fig. 4f–j, also plotted in Fig. 1d–h). When fitting the relative vertical displacement to the linear model, we avoid the peak caused by lake drainage. For consistency, we fit the linear model to the relative vertical displacement data, terminating at time $t_{end}$ (the right end of each plot in Supplementary Fig. 4), after the peak during $t_{end}/2 < t < t_{end}$ and before the peak during $-t_{end}/4 < t < 0$, as shown by the black lines in Supplementary Fig. 4a, b, e. For the 2011 data (Supplementary Fig. 4d) there are two uplift peaks after lake drainage, likely caused by additional water injected by nearby sources[2], thus the data during the first peak is avoided in the fitting of the linear model. The uplift data for Lake F (Supplementary Fig. 4c) is obtained from Fig. 6e,h in Doyle et al.[6]. Since Lake F data is not available for a longer time span, we simply subtract the data from its final vertical position (0.3 m and 0.2 m for stations NW and NE, respectively; Supplementary Fig. 4c). After detrending, the reference vertical displacements for all data sets are zero, as shown in Supplementary Fig. 4f–j.

Next, for the data in Supplementary Fig. 4f, g, i, j we applied the MATLAB function "rmoutliers" to detect and remove outliers, defined as observations more than 3 standard deviations away from the mean within a 4-day sliding window. To avoid removing the uplift peak at $t = 0$, the outliers were removed along the entire time series except for $|t| < 0.5$ days. The smoothed and detrended vertical displacements are shown in Supplementary Fig. 4f, g, i, j and Fig. 1d, e, g, h.

Finally, the smoothed and detrended vertical displacements $h(t)$ for $t > 0$ (Supplementary Fig. 4f–j) are then fitted to an exponential curve ($h(t) = h_i \exp(-t/t_{rel})$, where $h_i$ is the initial displacement at time zero; dashed curves in Supplementary Fig. 4f–j) to give the relaxation time $t_{rel}$ used for this study (vertical axis in Fig. 3a).

**South Lake 2009 drainage and volume.** Previous work has detailed multiple rapid drainages of North Lake[1,2,39,41]. Vertical and horizontal ice-sheet surface

displacements and lake drainage volumes have not been previously detailed for South Lake (68.57° N, 49.37° W) (Supplementary Fig. 5a). South Lake is located at 1050 m a.s.l. on the western margin of the Greenland Ice Sheet, roughly 22 km south of North Lake[41,42].

In 2009, South Lake rapidly drained on July 20, 2009, as indicated by uplift of GPS station SLSS (Supplementary Fig. 6b). No additional GPS were deployed in the vicinity of South Lake with a temporal sampling resolution adequate to observe the rapid drainage event in 2009. Ice-sheet vertical and horizontal displacements following the 2009 drainage of South Lake are most similar to the 2006 rapid drainage of North Lake (Supplementary Fig. 6a). SLSS station attains an uplift of 0.93 m over the initial 1.2 h of the drainage event, before relaxing to its pre-drainage elevation over the following ~14 h (Supplementary Fig. 6b). SLSS depicts a pre-drainage along-flow velocity of 141.7 m yr⁻¹, and a duration of 2.05 days is observed between the time of lake drainage and the time of along-flow displacement attaining the displacement predicted by pre-drainage along-flow velocities (Supplementary Fig. 6b).

The 30-m resolution MEaSUREs Greenland Ice Mapping Project (GIMP) Digital Elevation Model (DEM) from GeoEye and WorldView Imagery, Version 1[43,44] and Landsat 7 imagery were used to estimate the pre-drainage volume of South Lake. The last available image of South Lake in 2009 was taken on July 15, 2009, five days prior to South Lake's rapid drainage (Supplementary Fig. 5a). The lake margin on July 15, 2009 was mapped (Supplementary Fig. 5a) and used to compare to elevation contours of South Lake from the GIMP DEM (Supplementary Fig. 5b). The region of the GIMP DEM covering South Lake was created with a WorldView image taken on July 17, 2012[43], at which time South Lake basin did not host a lake, as South Lake drained on or before June 12, 2012 based on the Landsat image archive. Therefore, while there could be elevation changes between 2009 and 2012, the GIMP DEM over South Lake depicts a dry lake basin.

Based on a comparison between the mapped South Lake margin and GIMP DEM elevation contours, the South Lake margin elevation was estimated to be 1050.5 m a.s.l. (Supplementary Fig. 5b), yielding a lower bound on the lake volume estimate for South Lake of 0.029 km³, with maximum lake depths on the order of 10–15 m (Supplementary Fig. 5c). The GIMP DEM reports a 1σ error of ±1.27 m for every elevation estimate in the South Lake area (Supplementary Fig. 5b). A ±1σ error on the estimated lake margin elevation of 1050.5 m a.s.l. (Supplementary Fig. 5b) yields a lake volume error of 0.006 km³. This error estimate should presumably be a bit larger (~0.01 km³), given the GIMP DEM time stamp is in 2012, and the last pre-drainage image of South Lake was taken 4 days before the 2009 drainage. Following this methodology, we estimate the 2009 South Lake drainage volume to be 0.03 ± 0.01 km³.

**Uncertainties in Estimating Model Parameters**. The error bars of transmissivity $kh_0$ in Fig. 3a result from uncertainties in estimating (1) lake volume $V_{tot}$, (2) maximum blister height at the beginning of lake drainage $h_{max}$, and (3) blister radius $R$. The uncertainties associated with the three parameters are listed in Supplementary Table 2 and detailed below.

*Lake volume*. The lake volume $V_{tot}$ and its measurement errors for 2009 North Lake, 2010 Lake F, 2011 and 2012 North Lake are reported in Das et al.[1], Doyle et al.[6], Stevens et al.[2], respectively. The estimated uncertainty for 2006 South Lake volume is detailed in Methods. The lake volume values are listed in the table in Fig. 3 with error bars listed in Supplementary Table 2.

*Maximum blister height*. The maximum blister height $h_{max}$ for all drainage events are listed in the table in Fig. 3 with uncertainties listed in Supplementary Table 2. For 2011 and 2012 drainage events, $h_{max}$ are available from the Network Inversion Filter (NIF)[45] algorithm in Stevens et al. (2015)[2], which solves for the blister height based on surface displacement data from a network of 15 GPS stations around the blister at North Lake. We would expect the uncertainties in $h_{max}$ from NIF output to be on the order of magnitude of the uncertainties in the GPS elevation data (±0.05 m). The NIF calculation is not performed for 2006, 2009, and 2010 drainage events because there are not enough nearby GPS stations available to build an adequate network for the inversion, so $h_{max}$ is estimated by assuming that both (1) the initial blister shape [i.e., height to radius ratio ($Ar \equiv h_{max}/R = O(10^{-3})$)] and (2) the ratio of total water volume injected into the ice-bed interface (assumed to be the lake volume) to initial blister volume ($B \equiv V_{tot}/V_i = O(1)$) are the same for all blisters. Thus $V_{tot}/B = V_i = 2\pi\alpha h_{max}R^2 = 2\pi\alpha h_{max}^3/Ar^2 \propto h_{max}^3$, where $\alpha$ is a constant (equation (S.15)). We then estimate $h_{max}$ using the known lake volume $V_{tot}$ and the NIF-calculated blister height and lake volume from the 2011 drainage event, $h_{max}^{2011}$ and $V_{tot}^{2011}$, using $h_{max}/h_{max}^{2011} = (V_{tot}/V_{tot}^{2011})^{1/3}$. The error bar of this $h_{max}$ estimate is propagated from errors in lake volume measurements, as listed in the 4th column of Supplementary Table 2. Note that, even if we consider large variations of constants $B$ and $Ar$ between blisters (i.e., $B = O(0.1 - 10)$ and $Ar = O(10^{-4} - 10^{-2})$), the estimated $h_{max}$ will only vary within one order of magnitude.

*Blister radius*. Since blister volume is $V_i = 2\pi R^2 h_{max}$ (see equation (S.15)), the blister radius is estimated via $R = \sqrt{V_i/(2\pi\alpha h_{max})} = \sqrt{V_{tot}/(2\pi\alpha h_{max}B)}$

(estimated blister radii are listed in the table in Fig. 3) and the uncertainty associated with blister radius estimates is related to the uncertainties in estimating $V_{tot}$ and $h_{max}$ (5th column in Supplementary Table 2).

*Hydraulic transmissivity*. Finally, the errors associated with estimating the transmissivity $kh_0$ (last column in Supplementary Table 2) is propagated from errors in $V_{tot}, h_{max}$, and $R$. The error bar in $kh_0$ varies from 20% for the 2011 and 2012 drainage events to 40-50% for the 2006 and 2009 drainage events, due to larger uncertainties in both measuring lake volume and estimating maximum blister height.

*Numerical pre-factor f*. One benefit of dimensionless solutions is that the shape of uplift data $h(t)$ for a range of $V_{tot}$ and $kh_0$ can be compared with the theory on a dimensionless $h/h_i - t/t_c$ plot (Fig. 3c). Theoretically we expect a slight spread of dimensionless data on Fig. 3c because the analytical solution of blister height (red curve in Fig. 3c; Eq. (7)) depends not only on $h/h_i$ and $t/t_c$ but also $f = \alpha \ln[(B - \gamma)/C]$ (equation (S.26)). We estimate that $f$ varies between individual drainage events in the range $f = 0.5 - 0.8$, as shown in the last column in Table 1. In Fig. 3c we plot the solution using the value $f = 0.6$ averaged over all datasets (red dashed curve). The upper and lower bounds of the solutions calculated using $f = 0.5$ and $f = 0.8$, respectively, are illustrated by the black dashed lines in Fig. 3c.

*Systematic errors*. There are two constants that can cause systematic uncertainties: the Young's modulus of ice ($E \approx 10^{10}$ Pa) and the volume ratio between the lake volume and the initial blister volume ($B \equiv V_{tot}/V_i \approx 1.14$), estimated from the 2012 North Lake volume and the NIF-calculated initial blister volume. Since they cause systematic errors, variations in these parameters will not affect the collapse of the data in Fig. 3c, but will systematically shift the transmissivity value. If we consider an extreme parameter range, $E = O(10^9 - 10^{10})$ Pa and $B = 1 - 5$, the transmissivity $kh_0 \propto R^3/E \propto B^{-3/2}E^{-1}$ will be systematically multiplied by a factor in the range of 0.1 ~ 12.

**Transmissivity versus surface cumulative runoff**. The change of basal transmissivity between the early and mid-melt season is demonstrated in Fig. 4a. The x-axis is the transmissivity inferred from the five drainage events in 2006, 2009, 2010, 2011, and 2012 for the three lakes. The y-axis is the cumulative runoff from the first day of the year in which runoff occurs through the drainage date. The cumulative runoff shown by the y-position of the symbols in Fig. 4a is calculated from daily 11-km resolution RACMO runoff estimates[31] over the 11-km grid cell at locations of the GPS stations. The Lake F location is 67.01° N 48.74° W[6].

The difference in cumulative runoff estimates between RACMO[31] and MAR[34] in late July is large and reflected in the vertical error bars. The error bars of the cumulative runoff in Fig. 4a cover the range of values extrapolated from the regional climate MAR[34] (v3.5.2) model's monthly output averaged from 2006 to 2012 (y-positions of the symbols on the solid blue curve in Fig. 4b).

**Additional surface water injection near North Lake**. The minor uplift peaks observed at NL08 and NL09 ~6 days after the 2011 North Lake drainage could be related to additional water injected to the bed from basal or surface sources. The second uplift peak occurred on 2011/175 (day of year (DOY) 175 in 2011), roughly 6 days after the initial uplift peak on 2011/169. There are three LandSat-7 images available over this time window (Supplementary Fig. 8). From 2011/168 to 2011/170 (Supplementary Fig. 8a, b), North Lake is the only supraglacial feature to drain from view. Next, there is a 7-day gap in images with the next available LandSat-7 image on 2011/177 (Supplementary Fig. 8c). This image shows the drainage of two lakes immediately to the northwest of the northern extent of the GPS array. It is difficult to diagnose the drainage date and drainage style based on the images, but the regional bed topography (Supplementary Fig. 2b) suggests that water drained to the bed from these lakes could then flow beneath North Lake to avoid the basal ridge to the northwest. Based on this temporally sparse LandSat-7 imagery, two supraglacial lakes located 1–2 km from the northern extent of the GPS array are observed to drain within 1–8 days following the 2011 North Lake drainage (2011/169).

# Experimental methods

**Experimental setup**. We adhered a transparent elastic layer of polydimethylsiloxane (PDMS; Dow Corning Sylgard 184 Silicone Elastomer), which mimics ice, to a porous substrate (PDMS micropillar array), which serves as a simple model for the porous drainage system (Fig. 2a), using a double-sided adhesive tape (Drytac). When the fluid (glycerol dyed blue) was injected at the interface of the porous substrate and the adhered elastic layer[29], it first flowed within the porous substrate, then peeled apart the interface between the two layers, forming a blister. After injecting a total volume $V_{tot}$ of fluid (density $\rho$), we observed the time evolution of the fluid in the porous substrate and the blister

(Fig. 2b). Darker area indicates a larger blister thickness $H$, which relaxes with time, forcing a radial fluid flow into the porous substrate. The experimental parameters are listed in Table 1. In the experiments the elastic stresses ($\Delta p_e \approx EH/((1-\nu^2)R) \approx 10^4$ Pa) are the driving force for relaxation dynamics since they are much larger than the hydrostatic stresses of the blister fluid ($\rho g H \approx 1$ Pa), similar to the situation of a blister under the ice sheets.

The uncertainties associated with experimental parameters are listed in Supplementary Table 3.

**Fabrication of micropillar arrays.** A porous substrate made of micropillar arrays is used to model the porous drainage system. We designed and fabricated silicon molds for the micropillar arrays using standard photolithography methods. Each mold for the micropillar arrays is 7 cm in diameter and consists of circular wells on a hexagonal array with a porosity of 50%. Three variations of the micropillar array molds were designed with well diameters of 500 μm, 250 μm, and 125 μm and a well depth of 90 μm for each. Polydimethylsiloxane (PDMS) was cast onto the silicon molds using a crosslinker to elastomer ratio of 1 to 5 and cured to create PDMS micropillar arrays.

**Permeability measurement.** The permeability of each of the three micropillar arrays of depth $h_0 = 90$ μm was measured in a $w = 1$ cm wide by $L = 5$ cm long section of the micropillar array bonded to a glass slide. Water was injected into the device using a pressure control pump at a gauge pressure, $\Delta p$, and the flowrate, $Q$, was measured using a digital scale. From Darcy's law, $k = Q\mu L/(wh_0\Delta p)$, we determine the permeability, $k$, and its uncertainties of each micropillar array (listed in Table 1).

**Blister volume measurement.** A specified volume of glycerol was injected between the micropillar array and the overlying elastic sheet using a syringe pump. MATLAB was used for image processing to measure the area of the blister and the area of the fluid in the pores[29]. Time t = 0 is chosen to be the first image when the area of the blister has stopped increasing. By measuring the area of the fluid in the pores, the volume of fluid in the blister is calculated by subtracting the volume of the fluid in the pores from the total injected volume. The uncertainties in volume shown in Fig. 2c, d are from the error of the location of the boundaries of the blister and of the fluid in the pores and of the pore volume fraction $\phi$.

The uncertainty in the time scale (which includes elastic modulus, permeability, blister radius, and viscosity) is not large enough to be seen in Fig. 2d.

## Data availability

GPS Data for North Lake and South Lake are available at the UNAVCO repository (https://doi.org/10.7283/T55T3J80, https://doi.org/10.7283/T58K77VX, https://doi.org/10.7283/T54T6H4M, https://doi.org/10.7283/T5125RFN, https://doi.org/10.7283/T59K4915). All experimental data reported in this study are available at https://doi.org/10.34770/mx55-jz51.

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

## Acknowledgements

We thank J. Neufeld and D. Chandler for helpful discussions. C.-Y.L. and L.A.S thank Lamont-Doherty Earth Observatory for funding through the Lamont Postdoctoral Fellowships. D.L.C acknowledges support from the National Science Foundation (NSF) Graduate Research Fellowship. T.T.C. was supported by NSF's Office of Polar Programs (NSF-OPP) through OPP-1643970, the National Aeronautics and Space Administration (NASA) through NNX16AJ95G, and a grant from the Vetlesen Foundation. S.B.D. and M.D.B. acknowledge funding from NSF-OPP and NASA's Cryospheric Sciences Program through OPP-1838410, ARC-1023364, ARC-0520077, and NNX10AI30G. H.A.S. thanks the High Meadows Environmental Institute and the Carbon Mitigation Initiative at Princeton University. This publication was supported by the Princeton University Library Open Access Fund.

## Author contributions

All authors contributed to manuscript preparation. C.-Y.L led the project and the preparation of the manuscript. L.A.S, S.B.D., and M.D.B. supplied GPS uplift data, and with T.T.C. interpreted the transmissivity result. D.L.C. designed and conducted the experiments, and along with H.A.S. assisted with the model development.

## Competing interests

The authors declare no competing interests.
