## [Peer Review File · Nature Communications]

REVIEWER COMMENTS

Reviewer #1 (Remarks to the Author):

The manuscript by Lai et al. characterizes the subglacial transmissivity of the ice-bed interface beneath several different supraglacial lakes using a novel combination of theory, laboratory experiments, and observational data. Using multiple lake drainages in different locations and different years, the authors are able to loosely constrain two groupings of subglacial transmissivity beneath supraglacial lakes: a lower transmissivity when cumulative melt is low, and a higher transmissivity when cumulative melt is higher.

Overall, the manuscript logically and clearly presents a novel methodology for determining subglacial transmissivity beneath supraglacial lakes in Greenland. The choices made for the laboratory experiments are well justified and the link between the theoretical results and the GPS-derived surface elevations is clear. My main concerns are associated with the context of the observations within subglacial hydrology generally.

First, the results presented within this manuscript are described as indicative of seasonal evolution of subglacial transmissivity, yet there are really only two clusters of transmissivity measurements: a low transmissivity associated with low cumulative melt and a high transmissivity associated with higher cumulative melt. Two observations do not a season make – June and July are generally early to mid-melt season in this region of Greenland with melt starting around May and ending near the end of August. Framing these two measurements in terms of seasonal behavior is somewhat disingenuous. This same issue arises with regard to evolution (At least to my opinion, is the gradual change or development). Two measurements that are different indicate that there is clear change in subglacial transmissivity but arguing that it clearly indicates subglacial evolution seems a bit premature. This point is relatively minor in terms of revisions but does potentially alter the impact of the manuscript.

The second issue is how the subglacial system is framed. The manuscript clearly argues that the subglacial system beneath each of the lakes is inefficient during all measurements such that the observed transmissivity only applies to the distributed system (not to increased channelization or a combination of increased channelization and distributed system conductivity). This assumption is important because the evolution between an inefficient and efficient drainage configuration and its impact on large-scale subglacial conductivity and ice velocity is well established in the literature but evolving conductivity within the distributed system is relatively new (e.g., Andrews et al., 2014; Downs et al., 2018; Hoffman et al., 2016; Rada & Schoof, 2018).

I can't clearly argue that the subglacial systems beneath the lakes remain unchannelized and I don't think this text can clearly argue this either. The manuscript primarily relies on the work of

Chandler et al. (2013) and Dow et al. (2014) to make the case for a distributed only type drainage. Dow et al. (2014) use a limited channel only model and come to the conclusion that channels are unlikely to form during lake drainages, but may form with persistent meltwater inputs, this is further emphasized by recent idealized work by Poinar et al. (2019). The Chandler et al. dye tracing study does indicate that channelization (or at least water velocities associated with turbulent channelized flow) occurs at elevations of ~1000m, but that it occurs late in the melt season (August). Thus, the text concludes that because the lake drainages occur in June and July, the subglacial system beneath them must be channelized. However, the assumptions needed to come to this conclusion are generally weak because (1) The lakes are in a substantially different location than that of the dye tracing study – thus even if performed during the same years, the cumulative melt curves will be different and (2) Chandler et al. conclude that channelization near L41 occurs sometime between July 4 and August 1 (Chandler et al. (2013), Figure 2b), though the higher elevation locations seem to remain unchannelized. The lake drainages that had the higher transmissivity occurred in mid to late July, so the assumption – based nearly solely on Chandler et al. – that the subglacial system is purely unchannelized during the observations is not demonstrably true. As such, the work presented here cannot definitively characterize inefficient system transmissivity alone.

A suggestion: It may be more logical to emphasize that the subglacial system is a continuum between unchannelized and channelized flow, that purely turbulent and purely laminar flow are unlikely endmembers, and that new models are acknowledging this (e.g., Sommers et al., 2018). As such, large scale transmissivity would be more critical to identify than the transmissivity/conductivity of individual elements. Such emphasis would reduce the need to rely on assumed states of the subglacial system.

These two issues are not insurmountable; however, whether the impact of the presented results remains after revisions will need to be assessed. Minor comments can be found below.

Minor comments

18. While dye tracing studies do indicate evolution, so do a range of other study types. Dye tracing is also relatively limited in Greenland (only Chandler et al. & Cowton et al. to my knowledge).

20. Is the 1000 m based on Chandler et al.? And is it justified/known for the current study location or Greenland hydrology in general?

26. It would be relevant to mention the actual hydraulic transmissivities here.

27-28. This phrase seems to indicate that the lake drainage is the primary mechanism for the transmissivity evolution, which based on the information here, and past studies isn't necessarily the case. Consider a careful rephrasing.

38. Theory or theories?

44. It isn't paramount for characterizing sliding instead it is paramount in characterizing the relationship between seasonal perturbations and surface melt.

48. I think that Chandler et al. showed the transition to be around ~1200m or so, is 1000m

relevant widely, or just specifically?

52-54. Same as comment on lines 27-28. A better description of bulk transmissivity and how the method characterizes seasonal evolution using instantaneous drainage times at different elevations.

70. Be clear about the limitations of the lake drainages – the spatial spread of the lakes and the number of years covered in 5 lake drainage events.

78. larger or longer? Smaller or shorter?

87. Here it would be important to clarify the type of transmissivity – just to be safe – either bulk, distributed, or channelized.

89-99. See my primary remarks.

172. What field observations? The plotted elevation data in SI figure 7 makes the graphs difficult to interpret and panel b seems oversimplified relative to Hewitt et al. (2018), but there is clearly justification for using only porous flow. However, can porous media flow alone explain substantial, rapid ice sheet uplift downstream of lake drainages (Andrews et al. 2018; Hoffman et al. 2011). Or is a combination of both a turbulent sheet and distributed flow needed to truly explain lake drainages?

193. Does varying the media depth also affect the relaxation timescale?

215-218. if there is an argument for the presence of sediments, it would also be logical to point out that slight variations in till thickness could be an important control on the variation of transmissivity in different directions for the same lake – not just differences in conductivity.

230. Different times in different years. Cumulative melt isn't necessarily going to be the primary dictator of subglacial efficiency – variability is.

232. The two orders of magnitude is not over the melt season – it is between June and July, which is more like early to mid-melt season.

237-239. This sentence is poorly phrased, and the observations don't seem to reveal a seasonal evolution (or gradual change) due to their limitations. Instead, there is a shift to higher transmissivities that seems to be associated with the cumulative surface melt volume.

241. To my understanding, hydraulic conductivity in models is really poorly constrained, plus Flowers allows hydraulic conductivity to fluctuate in space and time, though in reality, layer thickness should only vary in space. This sentence may introduce unnecessary conflict without providing much to the discussion.

246-247. See main comment.

266. I am unconvinced that this method could provide an actual representation of seasonal evolution when lakes tend to 1. Drain once, 2. Drain in clusters, and 3. Have no clear pattern of drainage.

270. The presence of extensive surface-to-bed connections at higher elevations is still an area of debate.

273. What basal sliding law? I think this statement is overly general – how would changing transmissivity alter the timescale of sliding in response to meltwater accessing the bed? The following argument is considered well known and generally inferable from ice velocities.

276-278. This is a rather Greenland centric view – there have been a number of studies on till

properties in Antarctica and in the lab. There is an argument to be made here, certainly, but it seems a bit more nuanced than observations don't exist.

Figures

Figure 1. Panel e includes some GPS that are not plotted and not sure why a legend gets its own sub-panel label. This probably has to do with reporting in text, but if that is the case, it should be made clear in the caption and probably rearranged so that it is panel d.

Figure 3. Panel A provide the actual drainage date, not a qualitative assessment of drainage timing. Panels b and c are blurry and somewhat difficult to interpret plus the different plaster radii should be indicated by different lines.

SI Figure 7. See line note.

References

- Andrews, L. C., Catania, G. A., Hoffman, M. J., Gulley, J. D., Lüthi, M. P., Ryser, C., et al. (2014). Direct observations of evolving subglacial drainage beneath the Greenland Ice Sheet. *Nature*, 514(7520), 80–83. <https://doi.org/10.1038/nature13796>
- Chandler, D. M., Wadham, J. L., Lis, G. P., Cowton, T., Sole, A., Bartholomew, I., et al. (2013). Evolution of the subglacial drainage system beneath the Greenland Ice Sheet revealed by tracers. *Nature Geoscience*, 6(3), 195–198. <https://doi.org/10.1038/ngeo1737>
- Dow, C. F., Kulesa, B., Rutt, I. C., Doyle, S. H., & Hubbard, A. (2014). Upper bounds on subglacial channel development for interior regions of the Greenland ice sheet. *Journal of Glaciology*, 60(224), 1044–1052. <https://doi.org/10.3189/2014JoG14J093>
- Downs, J. Z., Johnson, J. V., Harper, J. T., Meierbachtol, T., & Werder, M. A. (2018). Dynamic hydraulic conductivity reconciles mismatch between modeled and observed winter subglacial water pressure. *Journal of Geophysical Research: Earth Surface*. <https://doi.org/10.1002/2017JF004522>
- Hoffman, M. J., Andrews, L. C., Price, S. A., Catania, G. A., Neumann, T. A., Lüthi, M. P., et al. (2016). Greenland subglacial drainage evolution regulated by weakly connected regions of the bed. *Nature Communications*, 7, 13903. <https://doi.org/10.1038/ncomms13903>
- Poinar, K., Dow, C. F., & Andrews, L. C. (2019). Long-Term Support of an Active Subglacial Hydrologic System in Southeast Greenland by Firn Aquifers. *Geophysical Research Letters*, 46(9), 4772–4781. <https://doi.org/10.1029/2019GL082786>
- Rada, C., & Schoof, C. (2018). Subglacial drainage characterization from eight years of continuous borehole data on a small glacier in the Yukon Territory, Canada. *The Cryosphere Discuss.*, 2018, 1–42. <https://doi.org/10.5194/tc-2017-270>
- Sommers, A., Rajaram, H., & Morlighem, M. (2018). SHAKTI: Subglacial Hydrology and Kinetic, Transient Interactions v1.0. *Geoscientific Model Development*, 11(7), 2955–2974. <https://doi.org/10.5194/gmd-11-2955-2018>

Reviewer #2 (Remarks to the Author):

The rapid drainage of supraglacial lakes injects substantial volumes of water to the bed of the Greenland ice sheet over a short time period. These rapid lake draining events can cause changes in subglacial hydrology and basal sliding, and therefore, lead to a transient ice-sheet acceleration.

In this manuscript, authors estimate local effective hydraulic transmissivity beneath rapidly draining supraglacial lakes (1000–1350 m a.s.l.) using a novel method that combines field observations from five draining events, a mathematical model, and laboratory experiments. To the best of my knowledge, this is the first study that provides estimates of local effective hydraulic transmissivity in the Greenland interior (where ice thickness exceeds 1000 m). However, perhaps even more significant contribution of this study is the novel method itself, which enables to infer transmissivity of the drainage system based on the surface uplift relaxation following rapid lake drainage events and the lake volume. As the authors mentioned, more observations of transmissivity are needed to constrain processes governing the sliding in the ice-sheet interior, and the presented novel model could be a useful tool for field-scientists to obtain more transmissivity estimates in the future. For these reasons, I recommend this manuscript to be accepted for publication after some major revisions.

Comments :

◆ Figures - general comments

- All figures should be submitted in higher resolution. While reading the manuscript, I found myself zooming in to magnify each figure of the panel in order to look at it in more detail, but that was often not possible due to the poor figure resolution. However, more modifications than just changing the resolution might be necessary. For example, Figure 1 has 9 panels which makes it quite busy and it might cause difficulties for people reading the manuscript in a printed/physical copy of the journal. Authors should consider splitting Figure 1 into two separate figures or ensuring that all panels are readable and clear even in printed format. Also, some panels look like they would normally be separate figures, but are grouped together here in order not to exceed the journal's limitation for the number of illustrations.

-Materials and Methods section relies heavily on figures in Supplementary materials, hence it requires the reader to frequently switch between the two which is affecting the readability of the paper.

- Color palette used throughout the manuscript (Figure 1 e), Figure 3, Figure 4 a)) is not color-blind friendly. If you are color-blind, in Figure 1e it is almost impossible to distinguish colors 4 and 9, similarly colors 3, 7, and 8 look almost the same. I would suggest to the authors to use one of the color-blind friendly palettes.

◆ Figure 1

- c) This panel shows locations of three draining lakes: North Lake, South Lake, and Lake F. Current description says: *'The locations of North Lake (NL), South Lake (SL), and Lake F (LF) are marked by the blue dots. The star at 1000 m a.s.l. is the North Lake location'*. Why is the North lake marked both with a blue dot and with a star? What do the other two blue stars at 1500 m (a.s.l) and 2000 m (a.s.l) represent? This question is answered in the description of Figure 4, however, this should be clarified much earlier, in the description of Figure 1.

- Colors used in d) don't look the same as colors in the legend e), this is problematic since colors used in d) are harder to discern than colors used in e). Similarly, the color in h) is slightly different than the corresponding color in e). However, this is less problematic since there is data from only one GPS station shown in that plot.

- f): *'The two uplift peaks following the 2011 drainage shown in panel f) likely result from additional water injection into the blister from nearby surface or basal sources'*

Were you able to detect the potential source of additional injected water? Was there a moulin nearby or a lake that has drained? Can you use satellite imagery to find out?

- d) and f): In this study, only data from 3 (2) GPS stations at North Lake were used in 2011 (2012). However, more than just 3 (2) stations were installed around the North Lake and used by Stevens et al (2015) for NIF. Were the chosen stations the only ones that recorded uplift? If that is the case, can we use the distance of other GPS stations from the lake and/or stations that have recorded uplift to constrain the blister radius (R) and how does that compare to the estimated blister radius R for North Lake?

- Line 294: *'GPS data for each station were processed individually 294 relative to the 30-s resolution Greenland GPS Network (GNET) KAGA base station located on 295 bedrock, 55 km from North Lake (32)'*.

More details about the processing of GPS data should be added in Methods. Since you use different datasets, did all GPS stations have the 30-s resolution or is this just the case for the

North Lake stations in 2011 and 2012? Is the data in Figure 1 displayed using 30s resolution or was it smoothed/averaged over a longer time-window? Has all GPS data been processed in the same way? Did you take into account the effect of multipath?

There seems to be a non-random scatter in the uplift data, clearly visible in panel f), that looks like spikes on top of the main trend. These spikes seem to appear daily (1 day period between spikes). We can observe similar behaviour (1 day period between spikes) for the station in panel i) as well. Interestingly, there might be a 12 hour period between spikes in panel d), but that is less conclusive.

What is the source of this noise? Can it be removed?

◆ Figure 2

- c) and d): There are error-bars shown in these panels, but related parameter uncertainties are not listed in S. Table 2.

◆ Figure 3

- b) It is difficult to discern different GPSstations' uplift data - SLSS, NE, and NW are almost impossible to see since other data is plotted over them, while NL07 2012 and NL08 2012 have very similar color (more similar than it appears in the legend in panel a)).

- c) It's even harder to discern different GPSstations' uplift data in this panel (NL08 2011 and NL09 2011 are dominating the plot). I understand the difficulty in plotting dimensionless data in the same plot since it collapses onto the same curve, but this panel/plot should be modified. Is it possible to reduce the scatter by preprocessing the data since that might make this panel more clear?

Laboratory experiments give this paper additional supporting evidence. It was fascinating to me that dimensionless experimental results (Figure 2d)) fall onto the same universal curve as the field data (Figure 3c)). I think that showing dimensionless experimental and field-data on the same plot could be a way of emphasizing this result. However, since panel c) is already too busy, authors could highlight it in the main text instead.

- a) I suggest adding the exact date of the lake drainage event to the Drainage time column in the legend. In this way, readers won't need to flip between the main paper and the supplementary material to check the dates in S. Table 1.

◆ Figure 4

- a) It is hard to see error bars for NL09 and NL08 (2011) since they overlap with the y-axis. Furthermore, all data points for the same lake drainage event have the same y-value while their transmissivity (x-) values don't vary much, making it hard to see all the data points. I suggest off-setting 0 on the x-axis slightly to the right so that NL09 and NL08 (2011) don't fall on the y-axis. Additionally, authors should consider either make markers slightly transparent or try using concentric circles of different sizes as their markers instead (e.g.

<https://i.stack.imgur.com/mi500.png>)

- a) Error bars: Horizontal (transmissivity) error bars are not visible for NL07, NL08 (2011,2012), and NL09 (2011,2012). Based on S. Table 2, I assume that the reason for that is that they are small enough to be covered by the marker. However, in S. Table 2 there are no estimates for cumulative runoff errors, please include them. How come there are no vertical error bars above the data points SLSS, NLBS, NL08, and NL09 (2011), and no vertical error bars below the data points NE and NW?

- a) '*Transmissivity inferred from five drainage events across three lakes as a function of cumulative runoff.*' If transmissivity is a function of cumulative runoff, transmissivity should be shown on the y-axis and cumulative runoff on the x-axis.

- d) '*...evaluated at the three locations (elevations) marked as stars in Fig. 1c.*' This was not clear from the description of Figure 1 where it was written '*The star at 1000 m a.s.l. is the North Lake location.*' Additional clarifications/descriptions of previous figures shouldn't be added in the later figures so this should be modified. Also, elevations are visible from contour lines, I'm not sure whether star markers are needed in Figure 1 to highlight them.

- b) Estimates of errors for cumulative runoff are not included. I suggest adding a shaded uncertainty to this plot.

◆ 229-243: Seasonal evolution of transmissivity under supraglacial lakes

- 230-234 Out of the observed five lake drainage events, three occurred at the same location (North Lake) at different times within the melt season, however, they happened over a span of 6 years (2006, 2011, and 2012). Supraglacial, englacial, and subglacial pathways can change from year to year, and the amount of available melt-water (daily and cumulative runoff) varies as well. Therefore, I think that it is not possible to claim that *the*

local transmissivity of the basal drainage system beneath North Lake can evolve by up to two orders of magnitude over the melt season. I agree that results suggest that transmissivity values appear to be lower earlier in the season and higher later in the season, which suggests that transmissivity likely evolves over the melt season. However, I don't think that we can infer how much it increases over the melt season by comparing transmissivity values from different melt-seasons (especially the ones 5-6 years apart). Do you assume that transmissivity beneath the North Lake stays the same from year to year at the beginning of the melt season? How does cumulative runoff vary from year to year? Can a lake drainage event cause formation of new moulin/conduit to glacier bed that weren't there the year before?

1269-272: *'Thus, over the coming decades as runoff increases and surface-to-bed meltwater pathways migrate inland to higher elevations (30), we expect an overall increase in transmissivity in the ice-sheet interior. Such an increase would likely impact the timescale of sliding in response to meltwater accessing the bed in ways not currently considered in the basal sliding law'*

However, Poinar et al (2015) claim: *' Thus, despite the observed inland migration of the ELA and surface melt in western Greenland, creation of new hydro fractures at high elevations is unlikely. Instead, most high-elevation meltwater likely will flow downhill via an extended network of surface streams and drain through existing moulins at lower elevations.'* Therefore, please, elaborate your claim about overall expected increase in transmissivity in the ice-sheet interior in more detail.

1 331-341: It looks like the authors meant to refer here to the supplementary Fig. 6, not the supplementary Fig. 5.

1442 Section 2.1: Experimental Setup

- Authors should consider including a video of experiments in Supplementary Materials. - Experiments for three different permeability values (k) were carried out. Did you test the repeatability of these experiments? How many times was each experiment repeated? Did the relaxation time and radius R vary? Did you calculate the uncertainty? Please, include more details on repeatability and estimating the uncertainty of laboratory experiments. There are error-bars shown in Figure 2.c-d, however, parameter uncertainties are not listed in S. Table 2.

► Supplementary Table 2: Parameter uncertainties

- There are N/As associated with h_{max} uncertainties for NL07, NL08 (both 2011 and 2012), NL09 (both 2011 and 2012). According to 378-381, for 2011 and 2012 drainage events, h_{max} are available from the Network Inversion Filter (NIF) algorithm in Stevens et al. (2015). How come there are no uncertainties associated with this output of the NIF algorithm?
- As mentioned before, uncertainties associated with laboratory experiments and with cumulative runoff estimates should be included in this table as well.

Editorial comments:

- 410-412 Authors use brackets for comments, although they usually use parentheses. Please, be consistent throughout the manuscript.
- It is possible that there are typos in the manuscript, however, this kind of editorial revisions are out of the scope of this review.

References:

Poinar, K. et al. Limits to future expansion of surface-melt-enhanced ice flow into the interior of western Greenland. Geophys. Res. Lett. 42, 1800–1807 (2015).

L. A. Stevens, M. D. Behn, J. J. McGuire, S. B. Das, I. Joughin, T. Herring, D. E. Shean, M. A. King, Greenland supraglacial lake drainages triggered by hydrologically induced basal slip. Nature. 525, 144 (2015).

Reviewer #3 (Remarks to the Author):

Review of 'Seasonally evolving hydraulic transmissivity beneath Greenland supraglacial lakes' by Amber Leeson

This paper primarily uses analytical modelling to simulate the evolution of a water-filled 'blister' at the base of an ice sheet. The intention is that this is analogous to a large quantity of water being delivered to the ice sheet base, and progressively draining away under the ice, when a supraglacial lake drains. In doing this, they come up with an estimate of the subglacial transmissivity in the region close to the

lake. The authors find that meltwater draining from surface lakes early in the melt season dissipates much slower than meltwater which enters the subglacial environment as a result of lake drainage later in the melt season. This is interesting because it suggests that a reconfiguration of the subglacial hydrological system occurs between early and late in the season, presumably from an inefficient distributed network to an efficient channelized system. As the authors point out - little is known about the configuration on the subglacial hydrological network this high up on the ice sheet so this is an interesting finding. The authors calibrate their results with field-based observations and with simple laboratory experiments. Together this is all very clever and nicely done.

The work is undoubtedly useful, and the paper is very well written. However it is mostly a description of the methods and less space is given to the application and interpretation of their new model. I would have liked to have seen this work extended to other areas – North East Greenland for example, or even simply other lakes in the South West, and used to develop a more comprehensive picture of the spatial and temporal evolution of subglacial drainage inland. At present the paper focuses on a very small area of a very large ice sheet.

I have a few specific comments below, but found the text to be very well-written and clear so I have no line by line comments:

1. The method is quite dependant on the GPS data, which is sparse. I wondered if the authors had given much thought to how important the temporal sampling of the GPS data is? Could this method be applied to satellite measurements of surface uplift for example, if (theoretically!) daily or weekly sampling was available?
2. In their model, the authors assume distributed subglacial drainage. This is well justified in the text but since this is highly uncertain – indeed also given the author's findings that the drainage system evolves - it would be interesting to contrast with a model that assumes channelized drainage.
3. The experimental portion of the paper is talked about in the results before it is properly introduced. E.g. on line 120 and in Figure 2. Consider rearranging, so that the experimental section comes after the GPS section and before the analytical model.
4. Some captions are somewhat verbose, suggest to keep them short and factual and move interpretation to the main text.
5. Not clear why a mix of MAR and RACMO is used in Figure 4 – I would have thought it would be better to pick one or average both in both plots?

Point-by-point response to reviewers for manuscript “Seasonally evolving hydraulic transmissivity beneath Greenland supraglacial lakes ”

Reviewer #1:

R1.1 The manuscript by Lai et al. characterizes the subglacial transmissivity of the ice-bed interface beneath several different supraglacial lakes using a novel combination of theory, laboratory experiments, and observational data. Using multiple lake drainages in different locations and different years, the authors are able to loosely constrain two groupings of subglacial transmissivity beneath supraglacial lakes: a lower transmissivity when cumulative melt is low, and a higher transmissivity when cumulative melt is higher.

Overall, the manuscript logically and clearly presents a novel methodology for determining subglacial transmissivity beneath supraglacial lakes in Greenland. The choices made for the laboratory experiments are well justified and the link between the theoretical results and the GPS-derived surface elevations is clear. My main concerns are associated with the context of the observations within subglacial hydrology generally.

Thank you for an insightful and constructive review, which helped to clarify important issues with regard to subglacial hydrology and to improve our manuscript. We are glad to hear that the reviewer finds our work novel and clearly presented. We have made substantial changes to the manuscript in response to your concerns and each of your comments are addressed below.

R1.2 First, the results presented within this manuscript are described as indicative of seasonal evolution of subglacial transmissivity, yet there are really only two clusters of transmissivity measurements: a low transmissivity associated with low cumulative melt and a high transmissivity associated with higher cumulative melt. Two observations do not a season make – June and July are generally early to mid-melt season in this region of Greenland with melt starting around May and ending near the end of August. Framing these two measurements in terms of seasonal behavior is somewhat disingenuous. This same issue arises with regard to evolution (At least to my opinion, is the gradual change or development). Two measurements that are different indicate that there is clear change in subglacial transmissivity but arguing that it clearly indicates subglacial evolution seems a bit premature. This point is relatively minor in terms of revisions but does potentially alter the impact of the manuscript.

We have edited the manuscript to clarify that we only have two clusters of data, rather than a continuous

evolution of data. Our result, rather than showing an evolution of the transmissivity, shows that the transmissivity likely evolved between early June and late July, due to the low transmissivity in June and high transmissivity in July. We have changed the following sentences to remove confusion.

We changed “*we compare the transmissivities for lakes draining at different times in the melt seasons*” to “*we compare the transmissivities for lakes draining at the early and mid melt seasons*”, changed “*Our results suggest the local transmissivity of the basal drainage system beneath North Lake can evolve by up to two orders of magnitude over the melt season*” to “*Our results suggest the local transmissivity of the basal drainage system beneath North Lake can change by up to two orders of magnitude between early June and late July.* ”, changed “*Our results suggest that local transmissivity beneath draining lakes at ~1000 m a.s.l. increases by up to two orders of magnitude throughout the melt season correlating to the volume meltwater moving through the system*” to “*Our results suggest that local transmissivity beneath North Lakes at ~1000 m a.s.l. differs by up to two orders of magnitude between June, 9, 2012 and July 29, 2006 as the cumulative surface melt volume differs by a factor of 25.*”, and changed “*The evolution of basal transmissivity over the course of the melt season is demonstrated in Fig. 4a.*” to “*The change of basal transmissivity between the early and mid melt season is demonstrated in Fig. 4a.*”

In addition, we made the following changes throughout the manuscript.

- 1) We replaced the word “transmissivity evolution” with “transmissivity change” because, as pointed out by the reviewer, evolution implies gradual continuous changes.
- 2) We replaced “over the course of the melt season” with “between early June and late July”
- 3) We replaced late-melt season with mid-melt season.

A similar challenge of a data gap was also visible in Chandler’s (2013) paper. Further away from the margin (41km away from the margin) their measured subglacial velocity shows a jump from low values in July to high values in August. How the basal hydrology evolves within the data gap (continuously, or by discrete jumps) is not well observed or known. Because of the lack of continuous evolving transmissivity data, we changed the paper title to “*Hydraulic transmissivity inferred from ice-sheet relaxation following Greenland supraglacial lake drainages*”, which better emphasizes the contribution of this work. As pointed out by Reviewer 2 “*this is the first study that provides estimates of local effective hydraulic transmissivity in the Greenland interior (where ice thickness exceeds 1000 m). However, perhaps even more significant contribution of this study is the novel method itself*”.

R1.3 The second issue is how the subglacial system is framed. The manuscript clearly argues that the subglacial system beneath each of the lakes is inefficient during all measurements such that the observed transmissivity only applies to the distributed system (not to increased channelization or a combination of increased channelization and distributed system conductivity). This assumption is important because the evolution between an inefficient and efficient drainage configuration and its impact on large-scale subglacial conductivity and ice velocity is well established in the literature but evolving conductivity within the distributed system is relatively new (e.g., Andrews et al., 2014; Downs et al., 2018; Hoffman et al., 2016; Rada & Schoof, 2018).

I can't clearly argue that the subglacial systems beneath the lakes remain unchannelized and I don't think this text can clearly argue this either. The manuscript primarily relies on the work of Chandler et al. (2013) and Dow et al. (2014) to make the case for a distributed only type drainage. Dow et al. (2014) use a limited channel only model and come to the conclusion that channels are unlikely to form during lake drainages, but may form with persistent meltwater inputs, this is further emphasized by recent idealized work by Poinar et al. (2019). The Chandler et al. dye tracing study does indicate that channelization (or at least water velocities associated with turbulent channelized flow) occurs at elevations of ~1000m, but that it occurs late in the melt season (August). Thus, the text concludes that because the lake drainages occur in June and July, the subglacial system beneath them must be channelized. However, the assumptions needed to come to this conclusion are generally weak because (1) The lakes are in a substantially different location than that of the dye tracing study – thus even if performed during the same years, the cumulative melt curves will be different and (2) Chandler et al. conclude that channelization near L41 occurs sometime between July 4 and August 1 (Chandler et al. (2013), Figure 2b), though the higher elevation locations seem to remain unchannelized. The lake drainages that had the higher transmissivity occurred in mid to late July, so the assumption – based nearly solely on Chandler et al. – that the subglacial system is purely unchannelized during the observations is not demonstrably true. As such, the work presented here cannot definitively characterize inefficient system transmissivity alone.

A suggestion: It may be more logical to emphasize that the subglacial system is a continuum between unchannelized and channelized flow, that purely turbulent and purely laminar flow are unlikely endmembers, and that new models are acknowledging this (e.g., Sommers et al., 2018). As such, large scale transmissivity would be more critical to identify than the transmissivity/conductivity of individual elements. Such emphasis would reduce the need to rely on assumed states of the subglacial system.

Thank you for the incisive comment and very helpful suggestion. In particular, it is an excellent suggestion to emphasize that the transmissivity in our study is a measure of a subglacial system with a

combination of localized flow through channels, distributed flow through sediments, thin films and linked cavities. We fully agree with you that large scale transmissivity would be more critical to identify than the transmissivity/conductivity of individual channels or a distributed network by itself.

Note that the Dow et al (2015) paper we referred to is different from the Dow et al (2014) paper mentioned by the reviewer. Nonetheless, we removed the sentence referring to Dow's (2015) model "*subglacial hydrological models indicate that water discharge remains in a distributed network after a lake drainage event (11)*" because this is poorly constrained by observations.

Our model does not rely on the assumption of a purely distributed sheet. As long as the channels are not big enough to switch the bulk subglacial discharge-potential gradient relationship under the lakes, Darcy's law can still be used to describe the subglacial system under the lakes, which is what we have assumed here. In fact, the increasing bulk transmissivity can be caused by many factors, including having more channels and weakly connected drainage (proposed by Hoffman et al (2016)) in the bulk subglacial system. Here we left the processes causing bulk transmissivity change for future work.

We added "*In reality, either a purely distributed system without any channels or a channelized system dominated by turbulent discharge are unlikely end members of the subglacial drainage system beneath the lakes. Therefore, instead of quantifying the efficiency of a purely distributed sheet or a purely turbulent channel, here we use transmissivity to characterize the bulk subglacial drainage system (valid within 2–6 km horizontal distance from the lake, as explained below), where a wide range of drainage features likely co-exist (e.g., distributed flows through porous sediment sheets (20), thin films (15–17) linked cavities (18, 19), localized flows through channels, and weakly connected systems (48)). A similar approach was developed by Sommers et al. (2012) (49). Because the horizontal extent of the water flow is much larger than the vertical extent, the bulk subglacial drainage system is treated as a continuum water sheet of effective depth h_0 , effective permeability k , and porosity ϕ (ratio between water-filled space to total space).*" We have also substantially rewritten the model section to acknowledge the use of transmissivity as a measure of efficiency for the bulk drainage system rather than a purely channelized or purely distributed system.

The following references are added in the manuscript: Dow et al, J. Glaciol (2014), Hoffman et al, Nature Communications (2016), and Sommers et al, Geoscientific Model Development (2018).

Minor comments

R1.4 18. While dye tracing studies do indicate evolution, so do a range of other study types. Dye tracing is also relatively limited in Greenland (only Chandler et al. & Cowton et al. to my knowledge).

We agree that other observation types (e.g., borehole and/or moulin water pressures combined with surface GPS observations) do indicate evolution of subglacial drainage efficiency seasonally (e.g., Andrews et al. (2014)) in Greenland; however, these study types only quantify evolution locally and do not directly observe hydraulic transmissivity. Dye-tracing studies directly measure drainage efficiency evolution along the drainage pathway (as we have stated in the sentence on L18), and these studies are most directly related to our quantification of subglacial drainage efficiency.

We elect to keep the sentence with a small change: *“Dye-tracing studies conducted in the western margin sector of the ice sheet have directly observed drainage efficiency to evolve seasonally along the drainage pathway.”*

Reference:

Andrews LC, Catania GA, Hoffman MJ, Gulley JD, Lüthi MP, Ryser C, Hawley RL and Neumann TA (2014) Direct observations of evolving subglacial drainage beneath the Greenland Ice Sheet. *Nature* 514(7520), 80–83 (doi:10.1038/nature13796)

R1.5 20. Is the 1000 m based on Chandler et al.? And is it justified/known for the current study location or Greenland hydrology in general?

The 1000 m elevation given is based on Chandler et al. (2013) and Cowton et al. (2013). This elevation is generalizable for western margin Greenland hydrology in general, as most all Greenland hydrology observations are from the southwest or central west sectors of the ice sheet. The lake drainages within our study fall within this region.

We have revised the previous sentence to make this sentence more clear: *“Dye-tracing studies conducted in the western margin sector of the ice sheet have directly observed drainage efficiency to evolve seasonally along drainage pathways. However, the local evolution of drainage systems further inland, where ice thicknesses exceed 1000 m, remains largely unknown (Chandler et al. (2013), Cowton et al. (2013)).”*

R1.6 26. It would be relevant to mention the actual hydraulic transmissivities here.

We added the actual values of transmissivities to the sentence *“...reveals hydraulic transmissivity and*

indicates a two-order-of-magnitude increase in subglacial transmissivity (from $0.9 \pm 0.3 \text{ mm}^3$ to $218 \pm 91.4 \text{ mm}^3$) as the melt season progresses”

R1.7 27-28. This phrase seems to indicate that the lake drainage is the primary mechanism for the transmissivity evolution, which based on the information here, and past studies isn't necessarily the case. Consider a careful rephrasing.

Great point. We have rewritten this sentence to indicate that it is the overall seasonal meltwater input (and not the lake drainages themselves) that drive the seasonal increase in transmissivity.

Edited sentence reads: *“This deflation obeys a universal relaxation law with a timescale that reveals hydraulic transmissivity and indicates a two-order-of-magnitude increase in subglacial transmissivity (from $0.9 \pm 0.3 \text{ mm}^3$ to $218 \pm 91.4 \text{ mm}^3$) as the melt season progresses, suggesting significant changes in basal hydrology beneath the lakes driven by seasonal meltwater input.”*

R1.8 38. Theory or theories?

Theories. There are two theory papers cited here, Schoof (2010) and Hewitt (2013), that modeled the evolution of subglacial hydrology.

R1.9 44. It isn't paramount for characterizing sliding instead it is paramount in characterizing the relationship between seasonal perturbations and surface melt.

Good point. We modified the sentence to *“Understanding the degree and the spatial extent of the drainage system transition is paramount in characterizing the relationship between seasonal ice flow perturbations and surface melt.”*

R1.10 48. I think that Chandler et al. showed the transition to be around ~1200m or so, is 1000m relevant widely, or just specifically?

We mean 1000 m to be relevant widely across the western margin of the ice sheet.

Sentences changed to: *“Field-based tracer studies conducted in the western margin region of the ice sheet (7,8) suggest a progressive channelization throughout the melt season, and indicate that a 5-fold increase in subglacial water speed occurs as a channelized system develops near the terminus (8). However, the effect of channelization tapers at higher elevations (roughly $>1000 \text{ m}$ above sea level [a.s.l.]) (11, 44), where evolution of local basal hydrology remains poorly constrained by observations.”*

R1.11 52-54. Same as comment on lines 27-28. A better description of bulk transmissivity and how the method characterizes seasonal evolution using instantaneous drainage times at different elevations.

Great point, again. We have rewritten this sentence to make clear that it is the overall seasonal meltwater input (and not the lake drainages themselves) that drive the seasonal increase in transmissivity.

Edited sentence reads: *“Here we calculate the local hydraulic transmissivity beneath rapidly draining supraglacial lakes (1000–1350 m a.s.l.) at their time of drainage using a new method combining field observations, a mathematical model, and laboratory experiments in order to quantify the seasonal changes of local hydraulic transmissivity driven by seasonal meltwater inputs.”*

R1.12 70. Be clear about the limitations of the lake drainages – the spatial spread of the lakes and the number of years covered in 5 lake drainage events.

We added the following information to the sentence. *“we characterize the relaxation timescale for five lake drainage events that occurred between 2006 and 2012 at three separate lakes located up to ~200 km apart along the western margin of the Greenland Ice Sheet (Fig. 1d-i; Supplementary Table 1)”*

R1.13 78. larger or longer? Smaller or shorter?

Fixed. It should be longer and shorter.

R1.14 87. Here it would be important to clarify the type of transmissivity – just to be safe – either bulk, distributed, or channelized.

See our response to R1.3. We substantially edited the model section to clarify the transmissivity is used to characterize the bulk drainage system.

R1.15 89-99. See my primary remarks.

We have rewritten this paragraph. See our response to R1.3.

R1.16 172. What field observations? The plotted elevation data in SI figure 7 makes the graphs difficult to interpret and panel b seems oversimplified relative to Hewitt et al. (2018), but there is clearly justification for using only porous flow.

The field observations refer to the GPS data presented in this paper. The plot colors for the GPS stations are equivalent across all main text and SI figures. We have added a legend in panel a that gives the GPS station name and year.

Hewitt et al (2018) predicts for an increasing-radius blister above a porous layer the blister height reduces with time as a power law with a time exponent of $-2/11$. Therefore the first step to compare their theory with our model is to check this $-2/11$ power, which is only visible on a loglog plot (SI figure 7b; although seems simplified, this is a standard analysis for power-law models). On the loglog plot, the blister height GPS data is neither a straight line nor following the $-2/11$ exponent (shown as slope on SI figure 7b). The Hewitt (2018) paper contains rigorous mathematical derivation for the system they considered, thus the discrepancy between model and data most likely arise from assumptions different from reality rather than any model errors. This is also a motivation for us to develop a different model that captures the dynamics of blister relaxation (SI figure 7a).

However, can porous media flow alone explain substantial, rapid ice sheet uplift downstream of lake drainages (Andrews et al. 2018; Hoffman et al. 2011). Or is a combination of both a turbulent sheet and distributed flow needed to truly explain lake drainages?

This is a good question. From our model analysis, in order to get the observed “exponential decay” of the ice-sheet uplift, the drainage mechanism in the bulk subglacial flow has to be Darcian. We do not see any reason for the bulk subglacial flow consisting both a distributed sheet and turbulent channels to disobey the Darcian flow law (porous media flow; horizontal pressure gradient linearly proportional to water speed) as long as the turbulent channels do not dominate the volumetric water flow rate of bulk subglacial system, as mentioned in our response to R1.3.

R1.17 193. Does varying the media depth also affect the relaxation timescale?

Another good question. In the experiment we only change k to verify the model (equation 4). In the experiment we can't change the media depth h_0 because of the fabrication process. But in equation 4 we can see that it is the permeability k times the depth h_0 (i.e. the transmissivity $k \times h_0$) that affects the relaxation time. Thus, if we can change the h_0 in the experiment we expect that to affect the relaxation timescale too. Note that, in the GPS data, we directly infer the changing transmissivity instead of separating the effect of changing permeability from changing depth of the bulk subglacial drainage system.

R1.18 215-218. if there is an argument for the presence of sediments, it would also be logical to point out that slight variations in till thickness could be an important control on the variation of transmissivity in different directions for the same lake – not just differences in conductivity.

We added “*The changes in transmissivity kh_0 could be caused by the evolution of not only the effective permeability k (or conductivity $K \equiv (kpw g)/\mu$) but also effective thickness h_0 of the bulk subglacial*

drainage system.”

R1.19 230. Different times in different years. Cumulative melt isn't necessarily going to be the primary dictator of subglacial efficiency – variability is.

We modified the sentence from “*we compare the transmissivities for lakes draining at different times in the melt season*” to “*we compare the transmissivities for lakes draining in the early and mid melt seasons.*” It is true cumulative melt isn't necessarily the primary indicator of subglacial efficiency. We added “*Note that our result does not imply transmissivity solely depends on cumulative surface runoff. Identifying the processes controlling the shift in transmissivity is left for future studies.*” in the main text to clarify that.

R1.20 232. The two orders of magnitude is not over the melt season – it is between June and July, which is more like early to mid-melt season.

Indeed. Thank you for the note. We replaced “late melt season” with “mid melt season” throughout the manuscript, and changed “*evolve by up to two orders of magnitude over the melt season*” to “*change by up to two orders of magnitude between early June and late July.*”

R1.21 237-239. This sentence is poorly phrased, and the observations don't seem to reveal a seasonal evolution (or gradual change) due to their limitations. Instead, there is a shift to higher transmissivities that seems to be associated with the cumulative surface melt volume.

To avoid the confusion, we change the sentence to “*Our results suggest that local transmissivity beneath North Lake at ~1000 m a.s.l. differs by up to two orders of magnitude between June, 9, 2012 and July 29, 2006 as the cumulative surface melt volume differs by a factor of 25 (Fig. 4a).*”

R1.22 241. To my understanding, hydraulic conductivity in models is really poorly constrained, plus Flowers allows hydraulic conductivity to fluctuate in space and time, though in reality, layer thickness should only vary in space. This sentence may introduce unnecessary conflict without providing much to the discussion.

The hydraulic conductivity value we used here is from Dow et al (2015) instead of Flowers and Clarke (2002). We remove the Flowers and Clarke reference from this sentence, which is a reference for the definition of conductivity, to avoid confusion. We agree the hydraulic conductivity in hydrology models is poorly constrained. Nonetheless we think a brief statement comparing our referred transmissivity with the conductivity in previous hydrology models can potentially be helpful for future efforts in

understanding the differences between hydrology models and our inferred transmissivity.

R1.23 246-247. See main comment.

We changed this sentence to “*We infer that local hydraulic transmissivity under North Lake (~1000 m a.s.l.) likely changes seasonally.*”

R1.24 266. I am unconvinced that this method could provide an actual representation of seasonal evolution when lakes tend to 1. Drain once, 2. Drain in clusters, and 3. Have no clear pattern of drainage. We removed the term “seasonal evolution” and the sentence now reads: “*Our method provides an approach to determine the magnitude of transmissivity of high-elevation regions using only observations of surface deformation and lake volume.*” Our current reported five drainage events (i.e. all available lake drainage GPS data in Greenland) gives us a limited view of transmissivity magnitude and seasonal evolution. However, there are hundreds of lake drainages across the ice sheet each year, and our model can be used to quantify transmissivity potentially under tens to hundreds of lakes across different sectors of the ice sheet. As noted by Reviewer 3, figuring out how to scale up the methodology we present here through field or remote sensing approaches is an area for future research.

R1.25 270. The presence of extensive surface-to-bed connections at higher elevations is still an area of debate.

Agreed! This is why we have written the sentence to indicate that the presence of surface-to-bed connections at higher elevations will migrate inland “over the coming decades”, and not at present. This is a forward-looking sentence that follows naturally from the prior sentence on projected runoff rates in 2100, and the sentence is well within the bounds of our concluding paragraph.

R1.26 273. What basal sliding law? I think this statement is overly general – how would changing transmissivity alter the timescale of sliding in response to meltwater accessing the bed? The following argument is considered well known and generally inferable from ice velocities.

We removed “*in ways not currently considered in the basal sliding law*” to avoid confusion.

It is well known that frictional sliding is typically sensitive to the pore fluid pressure. Therefore, throughout the melt season, transmissivity increases, pore fluid pressure decreases (effective pressure increases; increase basal drag). However, as transmissivity increases the ice-bed contact area also decreases because ice melts to form liquid (this may decrease basal drag). How increasing transmissivity translates to decreasing ice-bed contact is less well-known, but also impacts the basal drag. Therefore the

complete impacts of transmissivity on basal sliding drag seems to require further investigation. That said, the sentence contained too much information not directly related to the result so we removed this phrase.

R1.27 276-278. This is a rather Greenland centric view – there have been a number of studies on till properties in Antarctica and in the lab. There is an argument to be made here, certainly, but it seems a bit more nuanced than observations don't exist.

We have strong reservations against extending our findings from supraglacial lake drainages in Greenland to the Antarctic Ice Sheet, where neither seasonally evolving transmissivity or ice-sheet surface uplift following rapid supraglacial lake drainages on grounded ice have been taken/observed. We elect to keep this paper focused on Greenland Ice Sheet hydrology, though extending the work to compare till properties in Greenland, Antarctica, the lab, and other glacier beds could be an avenue for future work.

We replaced “*no observations of transmissivity exist for elevations above the five lakes investigated in this study*” with a clearer statement “*our observations of transmissivity are limited to the elevations of the five lakes investigated in this study, which are all located in the southwest to central west sectors of the Greenland Ice Sheet. Observations are needed at higher elevation rapidly draining lakes located across all sectors of the ice sheet to constrain processes governing the present and future evolution of sliding in the ice-sheet interior*”

R1.28 Figure 1. Panel e includes some GPS that are not plotted and not sure why a legend gets its own sub-panel label. This probably has to do with reporting in text, but if that is the case, it should be made clear in the caption and probably rearranged so that it is panel d.

We have revised this figure to take away the “e” designation to the legend. All nine GPS stations in the legend (formerly panel e) are plotted in panels d–h. Three stations are plotted in panel d, two stations are plotted in both panels e and f, and one station is plotted in both panels g and h.

R1.29 Figure 3. Panel A provide the actual drainage date, not a qualitative assessment of drainage timing. Panels b and c are blurry and somewhat difficult to interpret plus the different plaster radii should be indicated by different lines.

We have changed the qualitative assessment of drainage timing to the drainage date. We have resubmitted the figure at higher resolution.

R1.30 SI Figure 7. See line note.

We have improved the resolution of the figure.

References

- Andrews, L. C., Catania, G. A., Hoffman, M. J., Gulley, J. D., Lüthi, M. P., Ryser, C., et al. (2014). Direct observations of evolving subglacial drainage beneath the Greenland Ice Sheet. *Nature*, 514(7520), 80–83. <https://doi.org/10.1038/nature13796>
- Chandler, D. M., Wadham, J. L., Lis, G. P., Cowton, T., Sole, A., Bartholomew, I., et al. (2013). Evolution of the subglacial drainage system beneath the Greenland Ice Sheet revealed by tracers. *Nature Geoscience*, 6(3), 195–198. <https://doi.org/10.1038/ngeo1737>
- Dow, C. F., Kulesa, B., Rutt, I. C., Doyle, S. H., & Hubbard, A. (2014). Upper bounds on subglacial channel development for interior regions of the Greenland ice sheet. *Journal of Glaciology*, 60(224), 1044–1052. <https://doi.org/10.3189/2014JoG14J093>
- Downs, J. Z., Johnson, J. V., Harper, J. T., Meierbachtol, T., & Werder, M. A. (2018). Dynamic hydraulic conductivity reconciles mismatch between modeled and observed winter subglacial water pressure. *Journal of Geophysical Research: Earth Surface*. <https://doi.org/10.1002/2017JF004522>
- Hoffman, M. J., Andrews, L. C., Price, S. A., Catania, G. A., Neumann, T. A., Lüthi, M. P., et al. (2016). Greenland subglacial drainage evolution regulated by weakly connected regions of the bed. *Nature Communications*, 7, 13903. <https://doi.org/10.1038/ncomms13903>
- Poinar, K., Dow, C. F., & Andrews, L. C. (2019). Long-Term Support of an Active Subglacial Hydrologic System in Southeast Greenland by Firn Aquifers. *Geophysical Research Letters*, 46(9), 4772–4781. <https://doi.org/10.1029/2019GL082786>
- Rada, C., & Schoof, C. (2018). Subglacial drainage characterization from eight years of continuous borehole data on a small glacier in the Yukon Territory, Canada. *The Cryosphere Discuss.*, 2018, 1–42. <https://doi.org/10.5194/tc-2017-270>
- Sommers, A., Rajaram, H., & Morlighem, M. (2018). SHAKTI: Subglacial Hydrology and Kinetic, Transient Interactions v1.0. *Geoscientific Model Development*, 11(7), 2955–2974. <https://doi.org/10.5194/gmd-11-2955-2018>

Reviewer #2:

R2.1 The rapid drainage of supraglacial lakes injects substantial volumes of water to the bed of the Greenland ice sheet over a short time period. These rapid lake draining events can cause changes in subglacial hydrology and basal sliding, and therefore, lead to a transient ice-sheet acceleration.

In this manuscript, authors estimate local effective hydraulic transmissivity beneath rapidly draining supraglacial lakes (1000–1350 m a.s.l.) using a novel method that combines field observations from five draining events, a mathematical model, and laboratory experiments.

To the best of my knowledge, this is the first study that provides estimates of local effective hydraulic transmissivity in the Greenland interior (where ice thickness exceeds 1000 m). However, perhaps even more significant contribution of this study is the novel method itself, which enables to infer transmissivity of the drainage system based on the surface uplift relaxation following rapid lake drainage events and the lake volume. As the authors mentioned, more observations of transmissivity are needed to constrain processes governing the sliding in the ice-sheet interior, and the presented novel model could be a useful tool for field-scientists to obtain more transmissivity estimates in the future.

For these reasons, I recommend this manuscript to be accepted for publication after some major revisions.

We thank this reviewer for a very thorough and constructive review! We appreciate that the reviewer thinks “*even more significant contribution of this study is the novel method itself, which enables to infer transmissivity of the drainage system based on the surface uplift relaxation following rapid lake drainage events and the lake volume.*” We agree the method itself is a major contribution to Greenland subglacial hydrology where the measurement of effective transmissivity is difficult. Because of this and R2.19, we changed the paper title to “*Hydraulic transmissivity inferred from ice-sheet relaxation following Greenland supraglacial lake drainages*”, which better emphasizes the unique contribution of this work. In addition, we are glad that the reviewer finds the collapse of both the experimental and field data onto the same analytic solution fascinating, because we were surprised by this finding too.

► Figures - general comments

R2.2 All figures should be submitted in higher resolution. While reading the manuscript, I found myself zooming in to magnify each figure of the panel in order to look at it in more detail, but that was often not possible due to the poor figure resolution. However, more modifications than just changing the resolution might be necessary. For example, Figure 1 has 9 panels which makes it quite busy and it might cause difficulties for people reading the manuscript in a printed/physical copy of the journal. Authors should consider splitting Figure 1 into two separate figures or ensuring that all panels are readable and clear even

in printed format. Also, some panels look like they would normally be separate figures, but are grouped together here in order not to exceed the journal's limitation for the number of illustrations.

We have submitted higher resolution versions of all figures. We have made the panels larger in Fig. 1 to make it readable in a printed copy. We group the GPS data and the schematics (Fig1 a-b) in the same figure because the schematics shows the locations of the GPS stations that are marked in the table for the GPS data in Fig. 1.

R2.3 Materials and Methods section relies heavily on figures in Supplementary materials, hence it requires the reader to frequently switch between the two which is affecting the readability of the paper. We moved Supplementary Table 3 in the initial version to the revised main text (now Table 1). The rest of the supplemental items contain details rather than major results of the paper, so we kept them in the Supplementary Information. Because Nature Communication doesn't have Extended Figures, the figures mentioned in Methods are mostly in the Supplementary Information.

R2.4 Color palette used throughout the manuscript (Figure 1 e), Figure 3, Figure 4 a)) is not color-blind friendly. If you are color-blind, in Figure 1e it is almost impossible to distinguish colors 4 and 9, similarly colors 3, 7, and 8 look almost the same. I would suggest to the authors to use one of the color-blind friendly palettes.

Great point. We have fixed the colors of all GPS data to make it color-blind friendly following Paul Tol's muted color scheme. (<https://cran.r-project.org/web/packages/khroma/vignettes/tol.html>)

Reference:

Tol, Paul. 2018. "Colour Schemes." Technical note SRON/EPS/TN/09-002 3.1. SRON.

◆ Figure 1

R2.5 c) This panel shows locations of three draining lakes: North Lake, South Lake, and Lake F. Current description says: '*The locations of North Lake (NL), South Lake (SL), and Lake F (LF) are marked by the blue dots. The star at 1000 m a.s.l. is the North Lake location*'. Why is the North lake marked both with a blue dot and with a star? What do the other two blue stars at 1500 m (a.s.l) and 2000 m (a.s.l) represent? This question is answered in the description of Figure 4, however, this should be clarified much earlier, in the description of Figure 1.

Thanks for the suggestion! To reduce confusion, we remove the stars in the inset of Fig 1c and copy a map to Fig 4b, where we put the stars indicating the locations at 1500 m (a.s.l) and 2000 m (a.s.l) where

the cumulative runoff from MAR output is calculated.

R2.6 Colors used in d) don't look the same as colors in the legend e), this is problematic since colors used in d) are harder to discern than colors used in e). Similarly, the color in h) is slightly different than the corresponding color in e). However, this is less problematic since there is data from only one GPS station shown in that plot.

We have adjusted the color to make the color in GPS data consistent with that in the legend.

R2.7 f): *'The two uplift peaks following the 2011 drainage shown in panel f) likely result from additional water injection into the blister from nearby surface or basal sources'* Were you able to detect the potential source of additional injected water? Was there a moulin nearby or a lake that has drained? Can you use satellite imagery to find out?

There are two lakes that drains after the drainage of North Lake according to the LandSat imagery shown below (added as Supplementary Figure 8 in the revised manuscript). See the paragraph below we added in Methods.

"1.6 Additional surface water injection near North Lake

The minor uplift peaks observed at NL08 and NL09 ~6 days after the 2011 North Lake drainage could be related to additional water injected to the bed from basal or surface sources. The second uplift peak occurred on 2011/175 (day of year (DOY) 175 in 2011), roughly 6 days after the initial uplift peak on 2011/169. There are three LandSat-7 images available over this time window (Supplementary Fig. 8). From 2011/168 to 2011/170 (Supplementary Fig. 8a-b), North Lake is the only supraglacial feature to drain from view. Next, there is a 7-day gap in images with the next available LandSat-7 image on 2011/177 (Supplementary Fig. 8c). This image shows the drainage of two lakes immediately to the northwest of the northern extent of the GPS array. It is difficult to diagnose the drainage date and drainage style based on the images, but the regional bed topography (Supplementary Fig. 2b) suggests that water drained to the bed from these lakes could then flow beneath North Lake to avoid the basal ridge to the northwest. Based on this temporally sparse LandSat-7 imagery, two supraglacial lakes located 1–2 km from the northern extent of the GPS array are observed to drain within 1–8 days following the 2011 North Lake drainage (2011/ 169)."

We also moved the original sentence from the caption into the main text (in response to R3.6).

Supplementary Figure 8: Additional surface water injection near North Lake. North Lake (center of a), GPS stations (red triangles), and neighboring lakes in LandSat-7 images bracketing the North Lake drainage event on 2011/169 (day of year 169 in 2011).

R2.8 d) and f): In this study, only data from 3 (2) GPS stations at North Lake were used in 2011 (2012). However, more than just 3 (2) stations were installed around the North Lake and used by Stevens et al (2015) for NIF. Were the chosen stations the only ones that recorded uplift? If that is the case, can we use the distance of other GPS stations from the lake and/or stations that have recorded uplift to constrain the blister radius (R) and how does that compare to the estimated blister radius R for North Lake?

We were selective in our choice of which GPS stations to analyze because uplift data that will be useful for inferring transmissivity must (1) be at least ~10 days in length without temporal data gaps, (2) be largely free from additional surface melting events, and (3) be close enough to the lake to observe an uplift signal in the vertical positions. In general, at North Lake, stations >3 km from the lake-draining hydrofracture do not show uplift responses during the lake drainage (see figure below). So, yes, your suggestion to use the stations that did not record uplift to constrain the blister radius guided our initial selection of which GPS stations to analyze.

The blister radius we estimated (using a consistent method across all lake drainages) for the 2011 drainage event is 1.8 ± 1.0 km. An estimate of blister height has been inferred previously in Stevens et al (2015) (with the NIF algorithm) by inverting for basal cavity opening (which we take as the blister height) constrained by the spatially sparse GPS observations at the ice-sheet surface (triangles in figure below). In the NIF, the basal plane along which basal cavity opening can occur is discretized into 0.83 km x 0.83 km subfaults (grey squares in figure below), which is roughly the limit of horizontal resolution in basal cavity opening that we can observe given the locations of the GPS stations. Without assuming a

blister shape, a rough visual estimate of the blister radius from the NIF estimates of basal cavity opening suggests a radius of ~2km.

Blister height (meters) in 2011 at North Lake (presented as “basal cavity opening” in Stevens et al (2015)).

R2.9 Line 294: ‘GPS data for each station were processed individually 294 relative to the 30-s resolution Greenland GPS Network (GNET) KAGA base station located on 295 bedrock, 55 km from North Lake (32)’.

More details about the processing of GPS data should be added in Methods. Since you use different datasets, did all GPS stations have the 30-s resolution or is this just the case for the North Lake stations in 2011 and 2012? Is the data in Figure 1 displayed using 30s resolution or was it smoothed/averaged over a longer time-window? Has all GPS data been processed in the same way? Did you take into account the effect of multipath?

We have revised materials and Methods section 1.1 GPS to add more details about the GPS data and processing for North Lake and South Lake. We did not process the GPS position solutions for Lake F, but rather used the uplift data presented in Doyle et al. (2013) for Lake F, as noted in Section 1.2.

“1.1 North Lake and South Lake GPS Data

For the 2006, 2011, and 2012 North Lake and 2009 South Lake drainage events, continuous 30-s resolution GPS data collected by dual-frequency Trimble NetR9 receivers were processed with Track software (31), following the methodology previously presented for the same GPS data in Stevens et al. (2015) (6) and Stevens et al. (2016) (34). GPS data for each station were processed individually relative to the 30-s resolution Greenland GPS Network (GNET) KAGA base station located on bedrock, 55 km from North Lake (32). 1-σ error output from Track software for these data are on the order of ±2 cm in

the horizontal and ± 5 cm in the vertical across all stations and years (6). GPS data are archived at UNAVCO (see Data and materials availability).”

There seems to be a non-random scatter in the uplift data, clearly visible in panel f), that looks like spikes on top of the main trend. These spikes seem to appear daily (1 day period between spikes). We can observe similar behaviour (1 day period between spikes) for the station in panel i) as well. Interestingly, there might be a 12 hour period between spikes in panel d), but that is less conclusive. What is the source of this noise? Can it be removed?

Good point. The source of this 1-day period noise is likely due to daily changes in the orbit files, which results in larger noise at the day boundaries where orbit files change over. We removed the noise with a boxcar filter with a 4-day moving window along the timeseries. In the figure below, the smoothed and the original data for North Lake NL08 station is shown in gray and red, respectively. The GPS data in Fig. 1, 3, SI Fig. 4 (right panel) and SI Fig. 7 are all replaced with the smoothed version.

We added in Methods section 1.2:

“Next, for the data in Supplementary Fig. 4f,e,i,j we applied the MATLAB function “rmoutliers” to detect and remove outlier points, defined as those more than 3 standard deviations away from the mean within a 4-day sliding window. To avoid removing the uplift peak at $t=0$, the outliers were removed along the entire time series except for $|t|<0.5$ days. The smoothed and detrended vertical displacements are shown in Supplementary Fig. 4f,e,i,j.”

▶ Figure 2

R2.10 c) and d): There are error-bars shown in these panels, but related parameter uncertainties are not listed in S. Table 2.

The experimental error (Fig. 2c) is directly related to measurement technique for the blister volume. We added in Methods section 2.4 (experimental section) that “*The uncertainties in volume shown on Fig. 2c-d are from the error of the location of the boundaries of the blister and of the fluid in the pores and of the pore volume fraction ϕ .*” We added Supplemental Table 3 “*Parameter uncertainties of experimental data*” that summarizes the experimental parameter uncertainties, as summarized in our response to R2.23.

Supplemental Table 2 lists the uncertainties associated with field data. We clarify this by changing the title of Supplemental Table 2 to “*Parameter uncertainties of field data*”.

▶ Figure 3

R2.11 b) It is difficult to discern different GPSstations’ uplift data - SLSS, NE, and NW are almost impossible to see since other data is plotted over them, while NL07 2012 and NL08 2012 have very similar color (more similar than it appears in the legend in panel a)).

We updated the colors for all GPS stations and made the colors of NL07 2012 and NL08 2012 more distinctive.

c) It’s even harder to discern different GPSstations’ uplift data in this panel (NL08 2011 and NL09 2011 are dominating the plot). I understand the difficulty in plotting dimensionless data in the same plot since it collapses onto the same curve, but this panel/plot should be modified. Is it possible to reduce the scatter by preprocessing the data since that might make this panel more clear?

We re-processed all GPS data to systematically reduce the noise from all GPS the data (see response to R2.9). However, as the reviewer pointed out because the data collapse it would be difficult to see individual GPS data. For the purpose of showing each data clearly, we placed them in Fig. 1. However, to check how good the universal collapse is, we have to first put all data on the same graph (Fig. 3b; which inevitably causes overlapping datasets) to see how well the rescaled data merges onto the same curve (Fig. 3c) compared with Fig. 3b. The main point of Fig. 3b-c is to show a wide range of relaxation dynamics with different time scales (Fig 3b) collapse, which become indistinguishable when plotting on a dimensionless plot (Fig 3c).

R2.12 Laboratory experiments give this paper additional supporting evidence. It was fascinating to me that dimensionless experimental results (Figure 2d)) fall onto the same universal curve as the field data (Figure 3c)). I think that showing dimensionless experimental and field-data on the same plot could be a way of emphasizing this result. However, since panel c) is already too busy, authors could highlight it in the main text instead.

Thank you for the remark. We also find the collapse of the dimensionless experimental results (Figure 2d)) and the field data (Figure 3c)) onto the same universal curve fascinating. We added in the main text *“Notably, both the dimensionless field data (Fig. 3c) and dimensionless experimental data (Fig. 2d) fall onto the exponential solution (equation (6-7)), demonstrating the universality of the uplift relaxation dynamics.”*

R2.13 a) I suggest adding the exact date of the lake drainage event to the Drainage time column in the legend. In this way, readers won't need to flip between the main paper and the supplementary material to check the dates in S. Table 1.

We have changed the qualitative assessment of drainage timing to the exact drainage date.

▶ Figure 4

R2.14 a) It is hard to see error bars for NL09 and NL08 (2011) since they overlap with the y-axis. Furthermore, all data points for the same lake drainage event have the same y-value while their transmissivity (x-) values don't vary much, making it hard to see all the data points. I suggest off-setting 0 on the x-axis slightly to the right so that NL09 and NL08 (2011) don't fall on the y-axis. Additionally, authors should consider either make markers slightly transparent or try using concentric circles of different sizes as their markers instead (e.g. <https://i.stack.imgur.com/mi500.png>)

We off-set 0 on the x-axis slightly to the right. We also use different sizes of markers to distinguish the data points.

R2.15 a) Error bars: Horizontal (transmissivity) error bars are not visible for NL07, NL08 (2011,2012), and NL09 (2011,2012). Based on S. Table 2, I assume that the reason for that is that they are small enough to be covered by the marker.

That is correct! The horizontal error bars for 2011, 2012 data are too small and covered by the marker. You can see that the horizontal error for these events are visible on a loglog plot in Fig. 3a.

However, in S. Table 2 there are no estimates for cumulative runoff errors, please include them. How come there are no vertical error bars above the data points SLSS, NLBS, NL08, and NL09 (2011), and no vertical error bars below the data points NE and NW?

This is because the cumulative runoff of the symbols in Fig. 4a is from RACMO output, while the error bars come from the deviation of RACMO output from MAR model output. We decided to present the error as the difference between RACMO and MAR output because these differences are very large especially for the late-drainage events. There are no vertical error bars above the data points SLSS, NLBS, NL08, and NL09 (2011) simply because the cumulative runoff from MAR is smaller than that of RACMO. It is the other way around for NE and NW. The difference between MAR and RACMO is small enough that you don't see it for the 2012 events. We have explained this more clearly in Method section 1.5, as shown below:

“1.5 Transmissivity Versus Surface Cumulative Runoff

...The cumulative runoff shown by the y-position of the symbols in Fig. 4a is calculated from daily 11-km resolution RACMO runoff estimates (27) over the 11-km grid cell at locations of the GPS stations. The Lake F location is 67.01° N 48.74° W (4).

The difference in cumulative runoff estimates between RACMO (27) and MAR (29) in late July is large and reflected in the vertical error bars. The error bars of the cumulative runoff in Fig. 4a covers the range of values extrapolated from the regional climate MAR (29) (v3.5.2) model's monthly output averaged from 2006 to 2012 (y-positions of the symbols on the solid blue curve in Fig. 4b).”

R2.16 a) *‘Transmissivity inferred from five drainage events across three lakes as a function of cumulative runoff.’* If transmissivity is a function of cumulative runoff, transmissivity should be shown on the y-axis and cumulative runoff on the x-axis.

We changed it to *“Transmissivity inferred from five drainage events across three lakes and the corresponding cumulative runoff at the lake drainage time.”*

R2.17 b) *‘...evaluated at the three locations (elevations) marked as stars in Fig. 1c.’* This was not clear from the description of Figure 1 where it was written *‘The star at 1000 m a.s.l. is the North Lake location.’* Additional clarifications/descriptions of previous figures shouldn't be added in the later figures so this should be modified. Also, elevations are visible from contour lines, I'm not sure whether star markers are needed in Figure 1 to highlight them.

R2.18 b) Estimates of errors for cumulative runoff are not included. I suggest adding a shaded uncertainty to this plot.

The runoff in Fig. 4b is MAR model output. Because there are not many runoff measurements in Greenland, it's difficult to validate MAR runoff. To the best of our knowledge, the only comparison between in situ measurements and MAR runoff near North Lake is reported by Smith et al. (2017). The paper suggests that the models overestimate runoff in this region by ~20-40%. There aren't error bars for the 2100 runoff outputs because no observations exist and the MAR community does not publish errors for their model outputs. However, the point of Fig. 4b is to show that, according to MAR, in the future the runoff at higher elevations than the lakes can be larger than the current runoff at the lake elevations. The estimated runoff value does not affect the result of the paper (i.e., the transmissivity estimates).

References:

Smith, L.C., Yang, K., Pitcher, L.H., Overstreet, B.T., Chu, V.W., Rennermalm, Å.K., Ryan, J.C., Cooper, M.G., Gleason, C.J., Tedesco, M. and Jeyaratnam, J., 2017. Direct measurements of meltwater runoff on the Greenland ice sheet surface. *Proceedings of the National Academy of Sciences*, 114(50), pp.E10622-E10631.

◆ 229-243: Seasonal evolution of transmissivity under supraglacial lakes.

R2.19 230-234: Out of the observed five lake drainage events, three occurred at the same location (North Lake) at different times within the melt season, however, they happened over a span of 6 years (2006, 2011, and 2012). Supraglacial, englacial, and subglacial pathways can change from year to year, and the amount of available melt-water (daily and cumulative runoff) varies as well. Therefore, I think that it is not possible to claim that the local transmissivity of the basal drainage system beneath North Lake can evolve by up to two orders of magnitude over the melt season. I agree that results suggest that transmissivity values appear to be lower earlier in the season and higher later in the season, which suggests that transmissivity likely evolves over the melt season. However, I don't think that we can infer how much it increases over the melt season by comparing transmissivity values from different melt-seasons (especially the ones 5-6 years apart). Do you assume that transmissivity beneath the North Lake stays the same from year to year at the beginning of the melt season? How does cumulative runoff vary from year to year?

We changed “*We infer that local hydraulic transmissivity under North Lake (~1000 m a.s.l.) evolves changes seasonally.*” to “*We infer that local hydraulic transmissivity under North Lake (~1000 m a.s.l.) likely changes seasonally.*”, changed “*Our results show evidence that the drainage system evolves locally*

at the lake elevations (~1000 m a.s.l.). ” to “Our results implies that the drainage system likely evolves locally at the lake elevations (~1000 m a.s.l.). ”

Our statements that the transmissivity at the time of drainage for lake drainage events that occur at North lake but in different years indicate a seasonal evolution of transmissivity are based on the findings of two previous studies on North Lake basin annual and seasonal velocities and the regional subglacial drainage system (Stevens et al., 2016; 2018) and detailed field knowledge of the North Lake drainage events and surface-to-bed pathways (Das et al., 2006; Stevens et al., 2015). We assume that the transmissivity beneath the North Lake basin starts roughly at the same value at the beginning of the 2006, 2011, and 2012 melt seasons in comparison to the range in transmissivity that would occur over the melt season. While numerical modeling of the subglacial drainage system in the North Lake and wider region does show some lingering elevated water fluxes to persist for some model parameter choices (Stevens et al., 2018; Section 2.2.1), the seasonal range in model variables (sheet height, channel “height”) (Stevens et al., 2018; Fig. 7) is far greater than any year-to-year changes. Thus, it is reasonable to assume that lake drainages that occur only ~8–10% of the way into the melt season in 2011 and 2012 (Stevens et al., 2016; Figure S1, Table S2) would enter a subglacial drainage system that is quite different from the drainage system that occurs ~40 days later into the melt season (2006) (See Supplementary Table 1 of this manuscript). Moreover, in Stevens et al. (2016), cumulative annual runoff is given for the North Lake basin (Stevens et al., (2016); Fig. S4b), which, over the seven year time series of GPS data presented in that manuscript, shows high amounts of scatter but no statistically significant trend over time (Stevens et al. (2016); Fig. 3h). This suggests that the drainage system between 2006 and 2012 has not been consistently changing in one direction or another, but rather, experiencing a high variability in annual melt forcing from one year to the next.

Can a lake drainage event cause formation of new moulin/conduit to glacier bed that weren't there the year before?

We observe North Lake to access the same hydro-fracture and moulins (the moulins form along the hydro-fracture) in the 2006, 2011, and 2012 drainage events (Das et al., 2006; Stevens et al., 2015). The annual flow velocity in the North Lake basin is ~85 m yr⁻¹ (Stevens et al., (2016); Fig. 2a) and North Lake is ~2–3 km across along flowline, such that it would take 24–35 years for surface-to-bed features related to drainage (moulins, hydro-fracture scarps) to advect from one side of the lake basin to the other. From 2006 to 2012, features would have advected ~0.5 km. Thus, it doesn't seem likely that a new surface-to-bed pathway would be created in an entirely new area of the GPS array, keeping in mind that the GPS stations are already at least 1–2 km away from the lake margin.

To summarize, following from knowledge gained by the previous studies on North Lake, we assume that cumulative runoff varying from year to year does not have a substantial effect on the starting value of transmissivity beneath North Lake at the beginning of the melt season. By comparing the transmissivity estimates against cumulative runoff (Fig. 4 of this manuscript), we have focused our analysis on the seasonal change in melt forcing, which is the largest driver of subglacial drainage system change (e.g., Fig. 4 of Stevens et al., 2018) over the time period of the year when these lakes rapidly drain (i.e., the early- to mid-melt season).

References:

1. Stevens, Laura A., Mark D. Behn, Jeffrey J. McGuire, Sarah B. Das, Ian Joughin, Thomas Herring, David E. Shean, and Matt A. King. "Greenland supraglacial lake drainages triggered by hydrologically induced basal slip." *Nature* 522, no. 7554 (2015): 73-76.
2. Stevens, Laura A., Mark D. Behn, Sarah B. Das, Ian Joughin, Brice PY Noël, Michiel R. van den Broeke, and Thomas Herring. "Greenland Ice Sheet flow response to runoff variability." *Geophysical Research Letters* 43, no. 21 (2016): 11-295.
3. Stevens, L.A., Hewitt, I.J., Das, S.B. and Behn, M.D., 2018. Relationship between Greenland ice sheet surface speed and modeled effective pressure. *Journal of Geophysical Research: Earth Surface*, 123(9), pp.2258-2278.
4. Das, S.B., Joughin, I., Behn, M.D., Howat, I.M., King, M.A., Lizarralde, D. and Bhatia, M.P., 2008. Fracture propagation to the base of the Greenland Ice Sheet during supraglacial lake drainage. *Science*, 320(5877), pp.778-781.

R2.20 269-272: *‘Thus, over the coming decades as runoff increases and surface-to-bed meltwater pathways migrate inland to higher elevations (30), we expect an overall increase in transmissivity in the ice-sheet interior. Such an increase would likely impact the timescale of sliding in response to meltwater accessing the bed in ways not currently considered in the basal sliding law’*

However, Poinar et al (2015) claim: *‘ Thus, despite the observed inland migration of the ELA and surface melt in western Greenland, creation of new hydro fractures at high elevations is unlikely. Instead, most high-elevation meltwater likely will flow downhill via an extended network of surface streams and drain through existing moulins at lower elevations.’*

Therefore, please, elaborate your claim about overall expected increase in transmissivity in the ice-sheet interior in more detail.

The possibility and rate of the inland migration of surface-to-bed meltwater pathways is an open area of

research/debate at present (Christoffersen et al., (2018); MacFerrin et al. (2019)). We've edited the sentence to better reflect that debate, and to place our findings in the context of the most recent relevant work on observed and predicted changes to the percolation/upper ablation zone.

Edited sentence reads: “*Thus, over the coming decades as runoff increases (MacFerrin et al. (2019)) and if surface-to-bed meltwater pathways migrate inland to higher elevations (Poinar et al. (2015), Christoffersen et al. (2018)), we would expect an overall increase in transmissivity in the ice-sheet interior.*”

References:

1. Christoffersen P, Bougamont M, Hubbard A, Doyle SH, Grigsby S and Pettersson R (2018) Cascading lake drainage on the Greenland Ice Sheet triggered by tensile shock and fracture /704/106/125 /704/172/4081 /119 /134 article. *Nat. Commun.* **9**(1), 1–12 (doi:10.1038/s41467-018-03420-8)
2. MacFerrin MJ, Machguth H, van As D, Charalampidis C, Stevens CM, Heilig A, Vandecrux B, Langen PL, Mottram RH, Fettweis X, van den Broeke MR, Pfeffer WT, Moussavi M and Abdalati W (2019) Rapid expansion of Greenland's low-permeability ice slabs. *Nature* **573**, 403–407 (doi:10.1038/s41586-019-1550-3)

R2.21 331-341: It looks like the authors meant to refer here to the supplementary Fig. 6, not the supplementary Fig. 5.

Thank you for carefully catching the typo! It is fixed.

▶ 442 Section 2.1: Experimental Setup

R2.22 Authors should consider including a video of experiments in Supplementary Materials.

Thank you for the suggestion. We have uploaded the video as part of the supplementary information. We added a reference in the main text “*the blister thickness decreases and the blister radius remains unchanged (Supplementary video)*”

R2.23 Experiments for three different permeability values (k) were carried out. Did you test the repeatability of these experiments? How many times was each experiment repeated? Did the relaxation time and radius R vary? Did you calculate the uncertainty? Please, include more details on repeatability and estimating the uncertainty of laboratory experiments. There are error-bars shown in Figure 2.c-d, however, parameter uncertainties are not listed in S. Table 2.

The permeability (k) is the parameter that we directly control. The blister radius, initial volume, and total

volume have variation between each experiment. Experiments shown in Fig. 2 c-d are representative of more than 5 experiments that we conducted for each permeability.

Experimentally measured parameters:

Elastic Modulus: 217 ± 30 kPa

Permeability: $20 \pm 2, 52 \pm 9, 98 \pm 9$ μm^2

Glycerol Viscosity: 0.798 ± 0.005 Pa s

Pore fraction: 0.50 ± 0.01

Blister radius: ± 1 mm (from image processing limitations)

Fluid in pores radius: ± 0.1 mm (from image processing limitations)

These experimental parameters' uncertainties are listed in Supplemental Table 3 in the revised manuscript. We added in Method section 2 “*The uncertainties associated with experimental parameters are listed in Supplementary Table 3*” and “*The uncertainties in volume shown on Fig. 2c-d are from the error of the location of the boundaries of the blister and of the fluid in the pores and of the pore volume fraction ϕ . The uncertainty in the time scale (which includes elastic modulus, permeability, blister radius, and viscosity) is not large enough to be seen in Fig. 2d.*”

► Supplementary Table 2: Parameter uncertainties

R2.24 There are N/As associated with hmax uncertainties for NL07, NL08 (both 2011 and 2012), NL09 (both 2011 and 2012). According to 378-381, for 2011 and 2012 drainage events, hmax are available from the Network Inversion Filter (NIF) algorithm in Stevens et al. (2015). How come there are no uncertainties associated with this output of the NIF algorithm?

We mentioned “*For 2011 and 2012 drainage events, h_{max} are available from the Network Inversion Filter (NIF) (38) algorithm in Stevens et al. (2015) (6), which solves for the blister height based on surface displacement data from a network of 15 GPS stations around the blister at North Lake.*” The reason that we did not initially include any uncertainties associated with this output of the NIF algorithm is because the NIF algorithm does not calculate these uncertainties directly. The NIF works to minimize differences between deformation at the location of GPS receivers and displacement of an elastic halfspace surface (See Stevens et al. (2015) Extended Data Figs. 6 and 7). The uncertainties in h_{max} from NIF output would be sourced from uncertainties in the vertical height of the ice-sheet surface from GPS vertical data (± 0.05 m), the uncertainty in the choice of using an isotropic elastic halfspace as the model of the ice sheet for North Lake basin, the position of GPS stations at the ice sheet surface relative to the true center of the blister at the bed, and the uncertainties in the choice of NIF basal and vertical fault plane

locations and strike/dip orientations. Thus, we would expect that a lower bound on the errors for h_{\max} in the 2011 and 2012 drainage events to be on the order of magnitude of the uncertainties in the GPS elevation data (± 0.05 m).

We have revised Supplementary Table 2 to include the error of h_{\max} in the 2011 and 2012 drainage events, and propagate that error to R and kh_0 . You can see from Supplementary Table 2 that the final error of transmissivity kh_0 in 2011, 2012 is negligibly affected by including the error of h_{\max} . We added in Methods “*We would expect the uncertainties in h_{\max} from NIF output to be on the order of magnitude of the uncertainties in the GPS elevation data (± 0.05 m)*”. We updated the error bars of transmissivity in Fig.3a, 4a according to Supplementary Table 2.

R2.25 As mentioned before, uncertainties associated with laboratory experiments and with cumulative runoff estimates should be included in this table as well.

See our response to R2.10, R2.23.

◆ Editorial comments:

R2.26 410-412 Authors use brackets for comments, although they usually use parentheses. Please, be consistent throughout the manuscript.

We have changed the brackets for comments to parentheses consistently throughout the manuscript.

R2.27 It is possible that there are typos in the manuscript, however, this kind of editorial revisions are out of the scope of this review.

We thank the reviewer again for the very careful reading of our manuscript. We have reviewed the manuscript a few times to eliminate any typos we are aware of.

References:

Poinar, K. et al. Limits to future expansion of surface-melt-enhanced ice flow into the interior of western Greenland. *Geophys. Res. Lett.* 42, 1800–1807 (2015).

L. A. Stevens, M. D. Behn, J. J. McGuire, S. B. Das, I. Joughin, T. Herring, D. E. Shean, M. A. King, Greenland supraglacial lake drainages triggered by hydrologically induced basal slip. *Nature*. 525, 144 (2015).

Reviewer #3:

R3.1 This paper primarily uses analytical modelling to simulate the evolution of a water-filled ‘blister’ at the base of an ice sheet. The intention is that this is analogous to a large quantity of water being delivered to the ice sheet base, and progressively draining away under the ice, when a supraglacial lake drains. In doing this, they come up with an estimate of the subglacial transmissivity in the region close to the lake. The authors find that meltwater draining from surface lakes early in the melt season dissipates much slower than meltwater which enters the subglacial environment as a result of lake drainage later in the melt season. This is interesting because it suggests that a reconfiguration of the subglacial hydrological system occurs between early and late in the season, presumably from an inefficient distributed network to an efficient channelized system. As the authors point out - little is known about the configuration on the subglacial hydrological network this high up on the ice sheet so this is an interesting finding. The authors calibrate their results with field-based observations and with simple laboratory experiments. Together this is all very clever and nicely done.

Thank you for a very thoughtful review and suggestions for future work. We enjoyed thinking through your questions, which helped us to improve the manuscript and make our points more clear. We are also excited to hear that you find our paper undoubtedly useful and well written. The point-by-point responses to all of your comments are below.

R3.2 The work is undoubtedly useful, and the paper is very well written. However it is mostly a description of the methods and less space is given to the application and interpretation of their new model. I would have liked to have seen this work extended to other areas – North East Greenland for example, or even simply other lakes in the South West, and used to develop a more comprehensive picture of the spatial and temporal evolution of subglacial drainage inland. At present the paper focuses on a very small area of a very large ice sheet.

We agree that the paper focuses on a small area of the ice sheet. At this point, our method is limited to rapid supraglacial lake drainage events with in situ observations of surface uplift with hourly or finer temporal resolution. To our knowledge, we have presented all GPS data in previously published studies of rapidly draining lakes from GPS stations that are (1) close enough to the drained lake to observe an uplift response, (2) have hourly or finer temporal resolution, and, (3) have an uplift time series that extends long enough past the drainage event to fully observe the surface uplift relaxation back to pre-drainage elevations. We have not extended the work to the north, northeast, or southeast sectors of the ice sheet because there have not been in situ observations of rapid lake drainage in these regions. We did not include the data Tedesco et al. (2013) for Lake Ponting, for example, as the GPS stations are relatively far

away from Lake Pointing (compared to the length scale of other lakes in the immediate vicinity) (Tedesco et al. (2013) Figure 1), and the GPS time series does not extend long enough past the drainage event to fully observe the surface uplift relaxation back to pre-drainage elevations (Tedesco et al. (2013) Figure 3b). We are very excited to extend this method (which does need a lot of explanation in the text to be understood by observational glaciologists, theoretical glaciologists, and fluid dynamicists) to more lakes and other sectors of the ice sheet, and we expand on how this could be accomplished through future work in our response to your line comment R3.3 below.

Reference:

Tedesco M, Willis IC, Hoffman MJ, Banwell AF, Alexander P and Arnold NS (2013) Ice dynamic response to two modes of surface lake drainage on the Greenland ice sheet. *Environ. Res. Lett.* **8**(3), 34007.

I have a few specific comments below, but found the text to be very well-written and clear so I have no line by line comments:

R3.3 The method is quite dependent on the GPS data, which is sparse. I wondered if the authors had given much thought to how important the temporal sampling of the GPS data is?

Yes, given the timescale of the lake drainage events (hours) and surface uplift relaxation (1–10 days), the method is dependent on surface uplift (or surface elevation, more generally) data at hourly or finer timescales.

Could this method be applied to satellite measurements of surface uplift for example, if (theoretically!) daily or weekly sampling was available?

The method is dependent on GPS observations of surface uplift, as--to our knowledge--the temporal resolution of satellite observations is at most a couple of days and often longer than ~10 days. We are excited about the potential for the method to be extended to other areas of the ice sheet via additional field observations and/or the incorporation of remote sensing approaches. With all of the in situ studies of rapid lake drainages so far being located in the southwestern and central west sectors of the ice sheet, we think there would be much to learn about transmissivity by applying the method we present here to a larger population of lakes that covers a wider region (or all regions) of the ice sheet.

We have revised the final two sentences of the main text to indicate these future directions: *“To date, our observations of transmissivity are limited to the elevations of the five lakes investigated in this study, which are all located in the southwest to central west sectors of the Greenland Ice Sheet. Observations are needed at higher elevation rapidly draining lakes located across all sectors of the ice sheet to*

constrain processes governing the present and future evolution of sliding in the ice-sheet interior.”

R3.4 In their model, the authors assume distributed subglacial drainage. This is well justified in the text but since this is highly uncertain – indeed also given the author’s findings that the drainage system evolves - it would be interesting to contrast with a model that assumes channelized drainage.

Although we previously assumed a distributed sheet, we have revised the manuscript to clarify that our model does not rely on the assumption of a purely distributed sheet. As the reviewer points out the assumption of a purely distributed sheet is uncertain because channels could likely form. However, as long as the discharge in channels are not big enough (as supported by Dow’s model (2015)) to switch the bulk subglacial discharge-potential gradient relationship under the lakes, Darcy's law can still be used to describe the subglacial system under the lakes.

Below is the text added in the main text: *“In reality, either a purely distributed system without any channels or a channelized system dominant by turbulent discharge are unlikely end members of the subglacial drainage system beneath the lakes. Therefore, instead of quantifying the efficiency of a purely distributed sheet or a purely turbulent channel, here we use transmissivity to characterize the bulk subglacial drainage system (valid within 2–6 km horizontal distance from the lake, as explained later), where a wide range of drainage features likely co-exist (distributed flows through porous sediment sheets (20), thin films (15–17) and linked cavities (18, 19), localized flows through channels, and a weakly connected system (48)). Similar approach was developed by Sommers et al. (2012) (49). Because the horizontal extent of the water flows is much larger than the vertical extent, the bulk subglacial drainage system is treated as a continuum water sheet of effective depth h_0 , effective permeability k , and porosity ϕ (ratio between water-filled space to total space).”* We have also substantially rewritten the model section to acknowledge the use of transmissivity as a measure of efficiency for the bulk drainage system (as suggested by Reviewer 1) rather than a purely channelized or purely distributed system. In fact, the increasing bulk transmissivity can be caused by many factors, including having more channels and weakly connected drainage (proposed by Hoffman et al (2016)) in the bulk subglacial system. Here we left the processes causing bulk transmissivity change for future work.

R3.5 The experimental portion of the paper is talked about in the results before it is properly introduced. E.g. on line 120 and in Figure 2. Consider rearranging, so that the experimental section comes after the GPS section and before the analytical model.

Thank you for the suggestion. Currently the order of the sections is: GPS, analytical model, experiment, inferring transmissivity, discussion. After careful considerations we have decided to keep the current

order. Because the experiment serves as a justification for the analytical model, we think it would cause confusion if we jump into the details of the analogue experiment before the model section that properly describes the blister system.

R3.6 Some captions are somewhat verbose, suggest to keep them short and factual and move interpretation to the main text.

Agreed. We moved “*The two uplift peaks following the 2011 drainage shown in panel e likely result from additional water injection into the blister from nearby surface or basal sources.*” from Fig. 1 caption to the main text, and moved “*Since $t_c \propto R^3$ via equation (4), the relaxation time ($\sim 10^3$ seconds) of a laboratory-scale blister ($R \sim 10$ mm) is expected to be much shorter than that observed in the field.*” from Fig. 2 caption to the main text in the revised manuscript.

R3.7 Not clear why a mix of MAR and RACMO is used in Figure 4 – I would have thought it would be better to pick one or average both in both plots?

In Fig. 4a we used the cumulative runoff from RACMO. However, we want to show that the difference in cumulative runoff between RACMO and MAR is large, which is shown in the vertical error bars in Fig. 4a. We edited the Method section 1.5 to make this more clear.

“1.5 Transmissivity Versus Surface Cumulative Runoff

The evolution of basal transmissivity over the course of the melt season is demonstrated in Fig. 4a. The x-axis is the transmissivity inferred from the five drainage events in 2006, 2009, 2010, 2011, and 2012 among three lakes. The y-axis is the cumulative runoff from the first day of the year in which runoff occurs through the drainage date. The cumulative runoff shown by the y-position of the symbols in Fig. 4a is calculated from daily 11-km resolution RACMO runoff estimates (27) over the 11-km grid cell at locations of the GPS stations. The Lake F location is 67.01° N 48.74° W (4).

The difference in cumulative runoff estimates between RACMO (27) and MAR (29) in late July is large and reflected in the vertical error bars. The error bars of the cumulative runoff in Fig. 4a covers the values extrapolated from the regional climate MAR (29) (v3.5.2) model’s monthly output averaged from 2006 to 2012 (y-positions of the symbols on the solid blue curve in Fig. 4b).”

VIEWER COMMENTS

Reviewer #2 (Remarks to the Author):

Review: "Hydraulic transmissivity inferred from ice-sheet relaxation following Greenland supraglacial lake drainages" by C.-Y. Lai, L. A. Stevens, D. L. Chase, T. T. Creyts, M. D. Behn, S. B. Das, and H. A. Stone

All my concerns have been addressed in the revised version of the manuscript, all my questions have been carefully considered and the response/rebuttal was clear. I think the manuscript has been improved, particularly in terms of highlighting the novel method as the main impact of this manuscript, which is now clear from the new title.

The main concern that has been addressed was a claim that the seasonal evolution of subglacial transmissivity was observed which was also reflected in the title of the original manuscript. In my first review, I mentioned that I don't think that seasonal evolution is something we can conclude from the presented data (R2.19). This was addressed in more detail by reviewer 1 as well (R1.2, R1.24).

The authors have edited the manuscript to clarify that they only have two clusters of data, rather than a continuous evolution of data. Their result, rather than showing an evolution of the transmissivity, shows that the transmissivity likely evolved between early June and late July, due to the low transmissivity in June and high transmissivity in July. Because of the lack of continuous evolving transmissivity data, they changed the paper title to "Hydraulic transmissivity inferred from ice-sheet relaxation following Greenland supraglacial lake drainages", which better emphasizes the contribution of this work.

The fact that this is a method focused paper has been pointed out by reviewer 3 (R3.2) As I mentioned in my initial review, in my opinion, the biggest contribution of this manuscript is a novel method for inferring transmissivity from ice-sheet relaxation following supraglacial lake drainages. Using this method the authors were able to, to the best of my knowledge for the first time, estimate local effective hydraulic transmissivity in the Greenland interior. As reviewer 3 noted the current study is focused only on a small region, and I agree that it would be valuable to see this work extended to other areas of Greenland since as the authors have mentioned, more observations of transmissivity are needed to constrain processes governing the sliding in the ice-sheet interior and to better understand subglacial processes. However, I think that the larger-scale study is out of the scope of this paper, instead, it is something that we should strive for as a glaciology community, and the presented novel model will be a useful tool for other scientists to obtain more hydraulic transmissivity estimates in the future.

For these reasons, I recommend this manuscript be accepted for publication.

Reviewer #4 (Remarks to the Author):

Summary of manuscript

Lai et al. measure the subglacial hydraulic transmissivity (closely related to hydraulic conductivity) at three inland sites in western Greenland at specific times. They demonstrate that the surface response to basal water injection at different locations, dates/years, water volumes, and timescales (5 drainage events and 3 lab experiments) all follow an exponential decay function, which suits a simple model whose derivation is explained in detail. They infer the transmissivity from the decay rate constant (timescale of relaxation) and find great variation in transmissivity from lake to lake, which they interpret as variation over the course of the melt season (early June through late July). While their metric, hydraulic transmissivity (m^3), differs from the hydraulic conductivity used in

most subglacial models (m/s), they demonstrate that the values they attain are comparable to standard model parameter values (when a reasonable sheet/substrate thickness is used).

The work is a unique and strong combination of field observations, lab experiments, and a simple physically based model. The results fit well into our evolving understanding of the subglacial drainage system in Greenland. They add value by being a new and independent approach (flexure modeling) from typical (dye tracing or numeric models).

Specific points

The description of the bulk subglacial drainage system (lines 99-105), following reviewer comments, is much improved from its earlier version.

The figures are also now much more navigable. However, was the 1-day filter (described in the response to reviewers regarding orbit files) applied to the data in Figure 1e? The same 1-day periodic noise is still apparent.

Some discussion and a figure address the relationship between cumulative runoff and transmissivity (Fig. 4a) with speculation on future changes under RCP8.5 (Fig. 4b). The inclusion of the 1000 m RCP 8.5 projection in Fig. 4b misled me. It is far outside the range of what's been observed so far - by a factor of 3! - which is very grabbing on the figure. Fortunately, the authors don't speculate on this or use the 1000 m RCP 8.5 at all (they only use 1500 m). The authors should remove this curve from the figure, even the y-axes across panels a-b, and plot the lake drainage dates versus cumulative runoff on panel b. These edits would make it more convincing that the available data constrain the general forecasts of future basal transmissivity that the authors make in the discussion. I further suggest removing the circle and triangle markers from 4b (just plot the curves, there is nothing special about the first of each month) and plotting the lake data in blue (since they are all near 1000 m, although perhaps Lake F is closer to 1500 m, so orange). Finally, why do the runoff error bars only run in one direction, and why does that direction vary by lake? This should be explained in the caption, text, or supplement.

There should be some discussion on how the results affect interpretation/implications of other work. As it is now, the discussion rehashes the literature (lines 266-273), re-presents the results (one sentence, line 274-275), and then addresses future scenarios (second paragraph). One suggestion for this is that Downs et al. (2018) provides some precedent for the inferred linear relationship between runoff and basal conductivity (Fig. 4a). Downs prescribed this in GlaDS and found an improved match to borehole water pressure observations in the late melt season. (The Downs approach differs from the Lai approach by using raw runoff, not cumulative runoff.) Hewitt (2013) also commented on the apparent need for conductivity to change over the course of the year. Now, with this study, we have actual field evidence that this occurs.

Line by line points

Table 1 has a typo, "Porou"

Line 247 has a typo, "mm" should be mm^3

Line 258 From the values given on lines 256-7 and $\mu = 1\text{e-}3 \text{ Pa}\cdot\text{s}$ from Table 1, I get a range of h from 10^{-2} to 10^1 m (specifically 0.02 to 20 m). The low to mid end of this range still seems quite reasonable.

References

Downs, J. Z., J. V. Johnson, J. T. Harper, T. Meierbachtol, and M. A. Werder (2018), Dynamic Hydraulic Conductivity Reconciles Mismatch Between Modeled and Observed Winter Subglacial Water Pressure, *Journal of Geophysical Research: Earth Surface*, 123(4), 818–836, doi:10.1002/2017JF004522.

Point-by-point response to reviewers for manuscript “Seasonally evolving hydraulic transmissivity beneath Greenland supraglacial lakes ”

Reviewer #2:

Review: “Hydraulic transmissivity inferred from ice-sheet relaxation following Greenland supraglacial lake drainages” by C.-Y. Lai, L. A. Stevens, D. L. Chase, T. T. Creyts, M. D. Behn, S. B. Das, and H. A. Stone

All my concerns have been addressed in the revised version of the manuscript, all my questions have been carefully considered and the response/rebuttal was clear. I think the manuscript has been improved, particularly in terms of highlighting the novel method as the main impact of this manuscript, which is now clear from the new title.

The main concern that has been addressed was a claim that the seasonal evolution of subglacial transmissivity was observed which was also reflected in the title of the original manuscript. In my first review, I mentioned that I don’t think that seasonal evolution is something we can conclude from the presented data (R2.19). This was addressed in more detail by reviewer 1 as well (R1.2, R1.24).

The authors have edited the manuscript to clarify that they only have two clusters of data, rather than a continuous evolution of data. Their result, rather than showing an evolution of the transmissivity, shows that the transmissivity likely evolved between early June and late July, due to the low transmissivity in June and high transmissivity in July. Because of the lack of continuous evolving transmissivity data, they changed the paper title to “Hydraulic transmissivity inferred from ice-sheet relaxation following Greenland supraglacial lake drainages”, which better emphasizes the contribution of this work.

The fact that this is a method focused paper has been pointed out by reviewer 3 (R3.2)

As I mentioned in my initial review, in my opinion, the biggest contribution of this manuscript is a novel method for inferring transmissivity from ice-sheet relaxation following supraglacial lake drainages. Using this method the authors were able to, to the best of my knowledge for the first time, estimate local effective hydraulic transmissivity in the Greenland interior. As reviewer 3 noted the current study is focused only on a small region, and I agree that it would be valuable to see this work extended to other areas of Greenland since as the authors have mentioned, more observations of transmissivity are needed to constrain processes governing the sliding in the ice-sheet interior and to better understand subglacial processes. However, I think that the larger-scale study is out of the scope of this paper, instead, it is something that we should strive for as a glaciology community, and the presented novel model will be a useful tool for other scientists to obtain more hydraulic transmissivity estimates in the future.

For these reasons, I recommend this manuscript be accepted for publication.

We thank this reviewer for the thorough comments regarding our responses and the constructive suggestions from the first round which helped improve our manuscript!

Reviewer #4:

Summary of manuscript

Lai et al. measure the subglacial hydraulic transmissivity (closely related to hydraulic conductivity) at three inland sites in western Greenland at specific times. They demonstrate that the surface response to basal water injection at different locations, dates/years, water volumes, and timescales (5 drainage events and 3 lab experiments) all follow an exponential decay function, which suits a simple model whose derivation is explained in detail. They infer the transmissivity from the decay rate constant (timescale of relaxation) and find great variation in transmissivity from lake to lake, which they interpret as variation over the course of the melt season (early June through late July). While their metric, hydraulic transmissivity (m^3), differs from the hydraulic conductivity used in most subglacial models (m/s), they demonstrate that the values they attain are comparable to standard model parameter values (when a reasonable sheet/substrate thickness is used).

The work is a unique and strong combination of field observations, lab experiments, and a simple physically based model. The results fit well into our evolving understanding of the subglacial drainage system in Greenland. They add value by being a new and independent approach (flexure modeling) from typical (dye tracing or numeric models).

Specific points

The description of the bulk subglacial drainage system (lines 99-105), following reviewer comments, is much improved from its earlier version.

We thank the reviewer for the helpful comments. The Downs reference is particularly relevant because it demonstrates a nice connection between our result with the subglacial hydrological models.

The figures are also now much more navigable. However, was the 1-day filter (described in the response to reviewers regarding orbit files) applied to the data in Figure 1e? The same 1-day periodic noise is still apparent.

Yes, the 1-day filter was also applied to all GPS figures including Figure 1e, as mentioned in our response to reviewers and clarified in the methods. In fact Figure 1e before and after smoothing was plotted in our response to reviewers (round 1) to show the effect of the 1-day filter. The reason Figure 1e appears to be more noisy than other panels is largely because the time axis spans over longer timescale.

Some discussion and a figure address the relationship between cumulative runoff and transmissivity (Fig. 4a) with speculation on future changes under RCP8.5 (Fig. 4b). The inclusion of the 1000 m RCP 8.5 projection in Fig. 4b misled me. It is far outside the range of what's been observed so far - by a factor of 3! - which is very grabbing on the figure. Fortunately, the authors don't speculate on this or use the 1000 m RCP 8.5 at all (they only use 1500 m). The authors should remove this curve from the figure, even the y-axes across panels a-b, and plot the lake drainage dates versus cumulative runoff on panel b. These edits would make it more convincing that the available data constrain the general forecasts of future basal transmissivity that the authors make in the discussion. I further suggest removing the circle and triangle markers from 4b (just plot the curves, there is nothing special about the first of each month) and plotting the lake data in blue (since they are all near 1000 m, although perhaps Lake F is closer to 1500 m, so orange).

We made all changes suggested by the reviewer regarding Fig 4b: removing the 1000m RCP 8.5 projection, plotting the lake drainage dates versus cumulative runoff, and removing the circle and triangle markers from 4b. We also marked the lake locations on the map.

Finally, why do the runoff error bars only run in one direction, and why does that direction vary by lake? This should be explained in the caption, text, or supplement.

See our response to the round-1 review (R2.15), Method section 1.5, and the figure caption. This is because the error bar in Fig. 4a comes from the deviation of RACMO output from MAR model output while the symbols are cumulative runoff from RACMO output. We decided to present the error as the difference between RACMO and MAR output because they are very large especially for the late-drainage events. There are no vertical error bars above the data points SLSS, NLBS, NL08, and NL09 (2011) simply because the cumulative runoff from MAR is smaller than that of RACMO. It is another way around for NE and NW. The difference between MAR and RACMO is small enough that you don't see it for the 2012 events.

There should be some discussion on how the results affect interpretation/implications of other work. As it is now, the discussion rehashes the literature (lines 266-273), re-presents the results (one sentence, line 274-275), and then addresses future scenarios (second paragraph). One suggestion for this is that Downs et al. (2018) provides some precedent for the inferred linear relationship between runoff and basal conductivity (Fig. 4a). Downs prescribed this in GlaDS and found an improved match to borehole water pressure observations in the late melt season. (The Downs approach differs from the Lai approach by using raw runoff, not cumulative runoff.) Hewitt (2013) also commented on the apparent need for conductivity to change over the course of the year. Now, with this study, we have actual field evidence that this occurs.

We thank this reviewer for pointing us to the very relevant literature. We added in our manuscript: *“The seasonal change of hydraulic conductivity is important for reconciling mismatch between modeled and observed subglacial water pressure (49). Downs et al (2018) (49) included a simple linear relationship between melt input and hydraulic conductivity in a subglacial hydrological model and found an improved match between the modeled subglacial water pressure with the borehole observations in the late melt season. Our work provides observational evidence of the seasonal changes of hydraulic transmissivity.”*

Line by line points

Table 1 has a typo, "Porou"

Line 247 has a typo, "mm" should be mm³

Line 258 From the values given on lines 256-7 and $\mu = 1e-3 \text{ Pa}\cdot\text{s}$ from Table 1, I get a range of h_0 from 10^{-2} to 10^1 m (specifically 0.02 to 20 m). The low to mid end of this range still seems quite reasonable.

Thank you for the very careful read. We agree that h_0 ranges from 10^{-2} to 10^1 m and have fixed all typos you mentioned.

References

Downs, J. Z., J. V. Johnson, J. T. Harper, T. Meierbachtol, and M. A. Werder (2018), Dynamic Hydraulic Conductivity Reconciles Mismatch Between Modeled and Observed Winter Subglacial Water Pressure, *Journal of Geophysical Research: Earth Surface*, 123(4), 818–836, doi:10.1002/2017JF004522.

REVIEWERS' COMMENTS

Reviewer #4 (Remarks to the Author):

All of my comments have been addressed, save one: the y-axes for Figure 4a-b still do not match. However, their ranges are now similar (0-1200 meters and 0-2000 meters), which is improved from the previous version. The figure passes my inspection.

The manuscript will be a solid contribution that will influence the community's thinking about subglacial hydrology and lake drainage events, including the planning of future field work.

Reviewer #4:

All of my comments have been addressed, save one: the y-axes for Figure 4a-b still do not match. However, their ranges are now similar (0-1200 meters and 0-2000 meters), which is improved from the previous version. The figure passes my inspection.

The manuscript will be a solid contribution that will influence the community's thinking about subglacial hydrology and lake drainage events, including the planning of future field work.

Our response: We thank this reviewer for the thorough comments from the previous round of review which helped improve our manuscript.

Plotting Fig 4a and 4b on the same vertical scales (0-1200 meters) would hide some curves in Fig 4b (see figure below) and lead to confusion. We therefore believe it is best to keep the plot Fig 4a and 4b on their original vertical scales (0-1200 meters and 0-2000 meters), same as the previous resubmitted version so that the reader can see the complete curves in Fig. 4b. Note that the y-positions of the symbols in 4b are the same as either the upper or lower end of the y-error bars in 4a, as explained in the figure caption.